# Spectral embedding for dynamic networks with stability guarantees

**Ian Gallagher**
University of Bristol, UK
ian.gallagher@bristol.ac.uk

**Andrew Jones**
University of Bristol, UK
andrew.jones@bristol.ac.uk

**Patrick Rubin-Delanchy**
University of Bristol, UK
patrick.rubin-delanchy@bristol.ac.uk

## Abstract

We consider the problem of embedding a dynamic network, to obtain time-evolving vector representations of each node, which can then be used to describe changes in behaviour of individual nodes, communities, or the entire graph. Given this open-ended remit, we argue that two types of stability in the spatio-temporal positioning of nodes are desirable: to assign the same position, up to noise, to nodes behaving similarly at a given time (cross-sectional stability) and a constant position, up to noise, to a single node behaving similarly across different times (longitudinal stability). Similarity in behaviour is defined formally using notions of exchangeability under a dynamic latent position network model. By showing how this model can be recast as a multilayer random dot product graph, we demonstrate that unfolded adjacency spectral embedding satisfies both stability conditions. We also show how two alternative methods, omnibus and independent spectral embedding, alternately lack one or the other form of stability.

## 1   Introduction

Consider a dynamic network in which nodes come and go, change abruptly or evolve, alone or in communities, and form connections accordingly. Our goal is to find a vector representation, $\hat{\mathbf{Y}}_i^{(t)} \in \mathbb{R}^d$, for every node $i$ and time $t$, which could be used for a diversity of downstream analyses, such as clustering, time series analysis, classification, model selection and more. This problem is known as dynamic (or evolutionary) network (or graph) embedding, and a great number of scalable and empirically successful techniques have been put forward, with recent surveys by [47, 51]. We consider, among these, a subset about which it is reasonable to try to establish a certain statistical guarantee.

The novelty of this paper is *not* to propose a new procedure. Instead, it is to demonstrate that an existing procedure, unfolded adjacency spectral embedding (UASE) [17], has two important stability properties. Given a sequence of symmetric adjacency matrices $\mathbf{A}^{(1)}, \ldots, \mathbf{A}^{(T)} \in \{0,1\}^{n \times n}$, where $\mathbf{A}_{ij}^{(t)} = 1$ if nodes $i$ and $j$ form an edge at time $t$, UASE computes the rank $d$ matrix factorisation of $\mathbf{A} := (\mathbf{A}^{(1)} | \cdots | \mathbf{A}^{(T)})$ to obtain $\mathbf{A} \approx \hat{\mathbf{X}} \hat{\mathbf{Y}}^{\top}$ using the singular value decomposition (precise details later). The matrix $\hat{\mathbf{Y}} \in \mathbb{R}^{nT \times d}$ contains, as rows, the desired representations $\hat{\mathbf{Y}}_1^{(1)}, \ldots, \hat{\mathbf{Y}}_n^{(1)}, \ldots, \hat{\mathbf{Y}}_1^{(T)}, \ldots, \hat{\mathbf{Y}}_n^{(T)}$.

35th Conference on Neural Information Processing Systems (NeurIPS 2021).

The original motivation for UASE was to analyse multilayer (or multiplex) graphs under an appropriate extension of the random dot product graph model [17]. To evaluate UASE and other procedures on the task of dynamic network embedding, we instead consider the dynamic latent position model

$$\mathbf{A}_{ij}^{(t)} \overset{ind}{\sim} \text{Bernoulli}\left(f\left\{\mathbf{Z}_i^{(t)}, \mathbf{Z}_j^{(t)}\right\}\right), \tag{1}$$

for $1 \leq i < j \leq n$, $t \in [T]$, where $\mathbf{Z}_i^{(t)} \in \mathbb{R}^k$ represents the unknown position of node $i$ at time $t$, and $f : \mathbb{R}^k \times \mathbb{R}^k \to [0, 1]$ is a symmetric function.

This model is well-established [39, 22, 15, 16, 33, 23, 9, 43, 11, 10], with recent reviews by [20] and [45], and inference usually proceeds on the basis of a parametric model for $f$ (e.g. logistic in the latent position distance [39]) and for the dynamics of $\mathbf{Z}_i^{(t)}$ (e.g. a Markov process [39]). The model also includes the dynamic degree-corrected, mixed-membership and standard stochastic block models as special cases, which were studied in [48, 52, 14, 29, 50, 49, 31, 4, 32, 19].

To make statistical sense of UASE under this latent position model, we must somehow connect its output $\hat{\mathbf{Y}}_i^{(t)}$ to $\mathbf{Z}_i^{(t)}$. To this end we construct a canonical representative of $\mathbf{Z}_i^{(t)}$, denoted $\mathbf{Y}_i^{(t)}$, that UASE can be seen to estimate. Although we make some regularity assumptions on $f$ and the $\mathbf{Z}_i^{(t)}$, they are not modelled in an explicit, parametric way, and UASE can clearly be used in practice without having a specific model in mind. For example, we do not make a Markovian assumption on the evolution of $\mathbf{Z}_i^{(t)}$ and UASE can be used to uncover periodic behaviours.

The key purpose of imposing a dynamic latent position model is to allow us to put down certain embedding stability requirements. Using this framework, we can define precisely what we mean by two nodes, $i$ and $j$, behaving "similarly" at times $s$ and $t$ respectively. In such cases, ideally we would have $\hat{\mathbf{Y}}_i^{(s)} \approx \hat{\mathbf{Y}}_j^{(t)}$. Two special cases are: to assign the same position, up to noise, to nodes behaving similarly at a given time (cross-sectional stability) and a constant position, up to noise, to a single node behaving similarly across different times (longitudinal stability).

To achieve both cross-sectional and longitudinal stability is generally elusive. We show that two plausible alternatives, omnibus [26] and independent embedding of each $\mathbf{A}^{(t)}$, alternately exhibit one form of stability and not the other. More generally, we find existing procedures [6, 27, 40, 8, 54, 7, 28, 5, 4, 32, 37, 19] tend to trade one type of stability off against the other, e.g. via user-specified cost functions. As a side-note, it could be observed that omnibus embedding is not being evaluated on a task for which it was designed, since in the theory of [26] the graphs are identically distributed. Because the technique is so different from the others, we still feel it makes an interesting addition.

Our central contribution is to prove that UASE asymptotically provides both longitudinal and cross-sectional stability: for two nodes, $i$ and $j$, behaving similarly at times $s$ and $t$ respectively, we have $\hat{\mathbf{Y}}_i^{(s)} \approx \hat{\mathbf{Y}}_j^{(t)}$ and, moreover, $\hat{\mathbf{Y}}_i^{(s)}$ and $\hat{\mathbf{Y}}_j^{(t)}$ have asymptotically equal error distribution. We emphasise that these properties hold without requiring any sort of global stability — the network could vary wildly over time but certain nodes still stay fixed. In the asymptotic regime considered, we have $n \to \infty$, but $T$ fixed so that, for example, our results are relevant to the case of two large graphs. The alternative regime where $T \to \infty$ grows but $n$ is fixed is not easily handled by existing theory and to provide constant updates to UASE (or omnibus embedding) in a streaming context presents significant computational challenges.

The remainder of this article is structured as follows. Section 2 gives a pedagogical example demonstrating the cross-sectional and longitudinal stability of UASE in a two-step dynamic stochastic block model, while highlighting the instability of omnibus and independent spectral embedding. In Section 3, we prove a central limit theorem for UASE under a dynamic latent position model, and demonstrate that the distribution satisfies both stability conditions. In Section 4, we review the stability of other dynamic network embedding procedures. Section 5 presents an example of UASE applied to a dynamic network of social interactions in a French primary school, with a dynamic community detection example given in the main text and a further classification example provided in the Appendix. Section 6 concludes.

## 2   Motivating example

Suppose it is of interest to uncover dynamic community structure, including communities changing, merging or splitting. The dynamic stochastic block model [52, 50] provides a simple explicit model for this, in which two nodes connect at a certain point in time with probability only dependent on their current community membership. Suppose that at times 1 and 2, we have the following inter-community link probability matrices,

$$\mathbf{B}^{(1)} = \begin{pmatrix} 0.08 & 0.02 & 0.18 & 0.10 \\ 0.02 & 0.20 & 0.04 & 0.10 \\ 0.18 & 0.04 & 0.02 & 0.02 \\ 0.10 & 0.10 & 0.02 & 0.06 \end{pmatrix}, \quad \mathbf{B}^{(2)} = \begin{pmatrix} 0.16 & 0.16 & 0.04 & 0.10 \\ 0.16 & 0.16 & 0.04 & 0.10 \\ 0.04 & 0.04 & 0.09 & 0.02 \\ 0.10 & 0.10 & 0.02 & 0.06 \end{pmatrix}.$$

At time 1 there are four communities present, for example, a node of community 1 connects with a node of community 3 with probability 0.18. At time 2, this matrix changes so that communities 1 and 2 have merged, community 3 has moved, whereas community 4 is unchanged. For simplicity, in this example, the community membership of each node is fixed across time-steps.

We simulate a dynamic network from this model over these two time steps, on $n = 1000$ nodes, equally divided among the four communities and investigate the results of three embedding techniques: UASE, omnibus, and independent spectral embedding, displayed in Figure 1. Note that the different techniques produce embeddings of different dimensions; UASE embeds into $d = 4$ dimensions, omnibus embedding $\tilde{d} = 7$, while independent spectral embedding has $d_1 = 4$ and $d_2 = 3$. For visualisation, we show the leading two dimensions for each embedding. Given the dynamics described above, we contend that the following properties would be desirable:

1. *Cross-sectional stability*: The embeddings for communities 1 and 2 at time 2 are close.
2. *Longitudinal stability*: The embeddings for community 4 at times 1 and 2 are close.

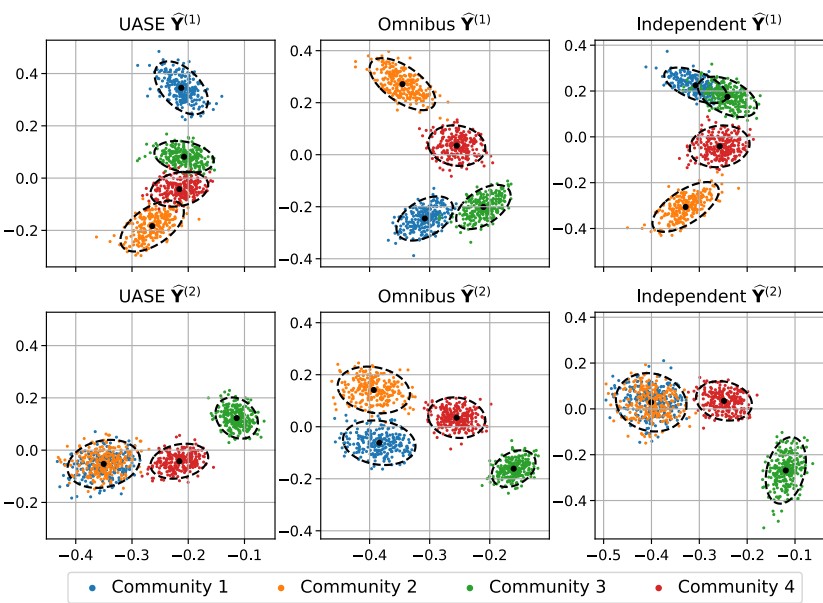

Figure 1: First two dimensions of the embeddings for the adjacency matrices $\mathbf{A}^{(1)}$ and $\mathbf{A}^{(2)}$ using three different techniques: UASE, omnibus and separate embedding. The points are coloured according to true community membership, with the black dots showing the fitted community centroid and the ellipses a fitted 95% level Gaussian contour.

Figure 1 illustrates that UASE has both properties: the blue and orange point clouds merge at time 2, and the red point cloud has the same location *and* shape over the two time points. On the other hand, omnibus embedding only has longitudinal stability (the blue and orange point clouds don't merge at

time 2), whereas independent spectral embedding only has cross-sectional stability (the red point cloud is at different locations over the two time points).

# 3 Theoretical background and results

We consider a sequence of random graphs distributed according to a dynamic latent position model, in which the latent position sequences $(\mathbf{Z}_i^{(t)})_{t \in [T]}$ (where we use the shorthand $[T] = \{1, \ldots, T\}$) are independent of each other, and identically distributed according to a joint distribution $\mathcal{F}$ on $\mathcal{Z}^T$, for a bounded subset $\mathcal{Z}$ of $\mathbb{R}^k$. We emphasise that for a fixed $i$ the latent positions $\mathbf{Z}_i^{(t)}$ and $\mathbf{Z}_i^{(t')}$ may have different distributions — we are imposing exchangeability over the nodes (invariance to node labelling), but not over time.

By extending the function $f$ to be zero outside of its domain of definition, we may view it as an element of $L^2(\mathbb{R}^k \times \mathbb{R}^k)$, and consequently may define a compact, self-adjoint operator $A$ on $L^2(\mathbb{R}^k)$ by setting

$$Ag(x) = \int_{\mathbb{R}^k} f(x, y)g(y)dy \tag{2}$$

for all $g \in L^2(\mathbb{R}^k)$. The operator $A$ has a—possibly infinite—sequence of non-zero eigenvalues $\lambda_1 \geq \lambda_2 \geq \ldots$ with corresponding orthonormal eigenfunctions $u_1, u_2, \ldots \in L^2(\mathbb{R}^k)$, such that $Au_j = \lambda_j u_j$ (for further details, see for example [35]). We shall make the simplifying assumption that $A$ has a *finite* number of non-zero eigenvalues, in which case $f$ admits a canonical eigendecomposition

$$f(\mathbf{x}, \mathbf{y}) = \sum_{i=1}^{D} \lambda_i u_i(\mathbf{x}) u_i(\mathbf{y}), \tag{3}$$

assumed to hold everywhere, where after relabelling we may assume that $|\lambda_1| \geq \ldots \geq |\lambda_D|$.

Several families of functions satisfy the finite rank assumption $D < \infty$, such as the multivariate polynomials [35]. Moreover, several existing statistical models can be written as dynamic latent position network models in which $D < \infty$, including the dynamic mixed membership, degree-corrected, and standard stochastic block models. To assume $D < \infty$, more generally, is tantamount to a claim that, for large $n$, $\mathbf{A}^{(t)}$ has 'low' approximate rank: an overwhelming proportion of its eigenvalues are close to zero. Large matrices with low approximate rank are routinely encountered across numerous disciplines, and the study [46] provides a hypothesis for this "puzzling" general observation. However, given a real network, we might reject the hypothesis that $f$ has low rank (we must then also reject any type of stochastic block model), for example on the basis of triangle counts [41]. In such a setting, we anticipate that UASE is still consistent and stable if $d$ is allowed to grow sufficiently slowly with $n$ and there exist asymptotic results for adjacency spectral embedding, for the single graph case, showing convergence in Wasserstein distance under assumptions on eigenvalue decay which can be related to the smoothness of $f$ [24]. However, even if we could obtain such results for UASE, they would not be as powerful as those we present here for the finite rank case, where we show uniform consistency with asymptotic Gaussian error.

**Theorem 1.** The adjacency matrices $\mathbf{A}^{(1)}, \ldots, \mathbf{A}^{(T)}$ are jointly distributed according to a multilayer random dot product graph model.

The multilayer random dot product graph model (MRDPG, see [17]) is a multi-graph analogue of the generalised random dot product graph [36], and is characterised by the existence of matrices $\mathbf{\Lambda}^{(t)} \in \mathbb{R}^{d \times d_t}$ and random matrices $\mathbf{X} \in \mathbb{R}^{n \times d}$ and $\mathbf{Y}^{(t)} \in \mathbb{R}^{n \times d_t}$ (generated according to some joint distribution $\mathcal{G}$) such that

$$\mathbf{A}_{ij}^{(t)} \overset{ind}{\sim} \text{Bernoulli}\big(\mathbf{X}_i \mathbf{\Lambda}^{(t)} \mathbf{Y}_j^{(t)\top}\big) \tag{4}$$

for each $t \in [T]$ and $i, j \in [n]$, where $\mathbf{X}_i$ and $\mathbf{Y}_j^{(t)}$ denote the $i$th and $j$th rows of $\mathbf{X}$ and $\mathbf{Y}^{(t)}$ respectively.

We refer the reader to the supplemental material for full details of the proof of Theorem 1, but note that given knowledge of the function $f$ and the underlying distribution $\mathcal{F}$ we can construct explicit

maps $\varphi : \mathbb{R}^{Tk} \to \mathbb{R}^d$ and $\varphi_t : \mathbb{R}^k \to \mathbb{R}^{d_t}$, producing row vectors, and matrices $\mathbf{\Lambda}^{(t)} \in \mathbb{R}^{d \times d_t}$ such that

$$f(\mathbf{Z}_i^{(t)}, \mathbf{Z}_j^{(t)}) = \mathbf{X}_i \mathbf{\Lambda}^{(t)} \mathbf{Y}_j^{(t)\top}, \tag{5}$$

where $\mathbf{X}_i = \varphi(\mathbf{Z}_i), \mathbf{Y}_j^{(t)} = \varphi_t(\mathbf{Z}_j^{(t)})$ and $\mathbf{Z}_i = (\mathbf{Z}_i^{(1)}|\cdots|\mathbf{Z}_i^{(T)}) \in \mathbb{R}^{Tk}$, for each $t \in [T]$ and $i, j \in [n]$. Consequently, each Gram matrix

$$\mathbf{P}^{(t)} = \big(f(\mathbf{Z}_i^{(t)}, \mathbf{Z}_j^{(t)})\big)_{i,j \in [n]} \tag{6}$$

admits a factorisation into a product of low-rank matrices, in which the matrix whose rows are the vectors $\varphi(\mathbf{Z}_i)$ appears as a common factor. The dimensions $d_t$ and $d$ are precisely the ranks of the matrices $\mathbf{P}^{(t)}$ and their concatenation $\mathbf{P} = (\mathbf{P}^{(1)}|\cdots|\mathbf{P}^{(T)})$ respectively. In the theory that follows we will assume that the dimension $d$ is known and fixed, whereas in practice we would usually have to estimate it (for example by using profile likelihood [55]).

We can extend our model to incorporate a range of sparsity regimes by scaling the function $f$ by a sparsity factor $\rho_n$, which we assume is either constant and equal to 1, or else tends to zero as $n$ grows, corresponding to dense and sparse regimes respectively. When this factor is present, the maps $\varphi$ and $\varphi_t$ are scaled by a factor of $\rho_n^{1/2}$.

Realising our model as an MRDPG allows us to make precise statements about the asymptotic behaviour of the point clouds obtained through UASE. In particular, it is the existence of a common factor in the decomposition of each of the Gram matrices $\mathbf{P}^{(t)}$ that gives rise to the stability properties previously demonstrated in the embeddings $\hat{\mathbf{Y}}^{(t)}$, as this matrix in a sense acts as an anchor for the individual point clouds.

In [17] UASE is used to obtain *two* multi-graph embeddings; the algorithm below retains only one — the 'dynamic' component.

---

**Algorithm 1** Unfolded adjacency spectral embedding for dynamic networks

---

    **input** symmetric adjacency matrices $\mathbf{A}^{(1)}, \ldots, \mathbf{A}^{(T)} \in \{0,1\}^{n \times n}$, embedding dimension $d$
1: Form the matrix $\mathbf{A} = (\mathbf{A}^{(1)}|\cdots|\mathbf{A}^{(T)}) \in \{0,1\}^{n \times Tn}$ (column concatenation)
2: Compute the truncated singular value decomposition $\mathbf{U_A} \mathbf{\Sigma_A} \mathbf{V_A}^\top$ of $\mathbf{A}$, where $\mathbf{\Sigma_A}$ contains the $d$ largest singular values of $\mathbf{A}$, and $\mathbf{U_A}, \mathbf{V_A}$ the corresponding left and right singular vectors
3: Compute the right embedding $\hat{\mathbf{Y}} = \mathbf{V_A} \mathbf{\Sigma_A}^{1/2} \in \mathbb{R}^{Tn \times d}$ and create sub-embeddings $\hat{\mathbf{Y}}^{(t)} \in \mathbb{R}^{n \times d}$ where $\hat{\mathbf{Y}} = (\hat{\mathbf{Y}}^{(1)}; \ldots; \hat{\mathbf{Y}}^{(T)})$ (row concatenation)
    **return** node embeddings for each time period $\hat{\mathbf{Y}}^{(1)}, \ldots, \hat{\mathbf{Y}}^{(T)}$

---

Replacing $\mathbf{A}$ with $\mathbf{P}$ in this construction yields the *noise-free* embeddings $\tilde{\mathbf{Y}}^{(t)}$, whose rows are known to be linear transformations of the vectors $\varphi_t(\mathbf{Z}_i^{(t)})$ (see the supplemental material for further details). A desirable property of UASE is that since the matrices $\mathbf{A}^{(t)}$ follow an MRDPG model there exist known asymptotic distributional results for the embeddings $\hat{\mathbf{Y}}^{(t)}$. In order to ensure the validity of these results, we will make the assumption that the sparsity factor $\rho_n$ satisfies $\rho_n = \omega(n^{-1} \log^c(n))$ for some universal constant $c > 1$.

In the results to follow, UASE is shown to be consistent and stable in a uniform sense, i.e. the maximum error of any position estimate goes to zero, under the assumption that the average network degree grows polylogarithmically in $n$. To achieve this for less than logarithmic growth would be impossible, by any algorithm, because it would violate the information-theoretic sparsity limit for perfect community recovery under the stochastic block model [1]. In practice this means that the embedding will have high variance when the graphs have few edges. Several approaches have been proposed to make single graph embeddings more robust (e.g. to sparsity or heterogeneous degrees), such as based on the regularised Laplacian [3] or non-backtracking matrices [21], but it is an open and interesting question how to extend them to the dynamic graph setting to achieve both cross-sectional and longitudinal stability.

The first of our distributional results states that after applying an orthogonal transformation—which leaves the structure of the resulting point cloud intact—the embedded points $\hat{\mathbf{Y}}_i^{(t)}$ converge in the Euclidean norm to the noise-free embedded points $\tilde{\mathbf{Y}}_i^{(t)}$ as the graph size grows:

**Proposition 2.** There exists a sequence of orthogonal matrices $\tilde{\mathbf{W}} \in O(d)$ such that

$$\max_{i \in \{1,\dots,n\}} \left\| \hat{\mathbf{Y}}_i^{(t)} \tilde{\mathbf{W}} - \tilde{\mathbf{Y}}_i^{(t)} \right\| = O\left( \frac{\log^{1/2}(n)}{\rho_n^{1/2} n^{1/2}} \right) \tag{7}$$

with high probability for each $t$.

The matrices $\tilde{\mathbf{W}}$ are unidentifiable in practice, but are defined to be the solution to the one-mode orthogonal Procrustes problem

$$\tilde{\mathbf{W}} = \underset{\mathbf{Q} \in O(d)}{\arg\min} \| \mathbf{U_A Q} - \mathbf{U_P} \|_F^2 + \| \mathbf{V_A Q} - \mathbf{V_P} \|_F^2, \tag{8}$$

where $\|\cdot\|_F^2$ denotes the Frobenius norm. We emphasise that the matrix $\tilde{\mathbf{W}}$ is *the same* for each embedding, and so similar behaviour in the noise-free embeddings $\tilde{\mathbf{Y}}^{(t)}$ is captured by UASE.

Our second result states that after applying a *second* orthogonal transformation the above error converges in distribution to a fixed multivariate Gaussian distribution:

**Proposition 3.** Let $\zeta = (\zeta_1 | \cdots | \zeta_T) \sim \mathcal{F}$, and for $\mathbf{z} \in \mathcal{Z}$ define

$$\mathbf{\Sigma}_t(\mathbf{z}) = \begin{cases} \mathbb{E}_\zeta \Big[ f(\mathbf{z}, \zeta_t) \big( 1 - f(\mathbf{z}, \zeta_t) \big) \cdot \varphi(\zeta)^\top \varphi(\zeta) \Big] & \text{if } \rho_n = 1 \\ \mathbb{E}_\zeta \Big[ f(\mathbf{z}, \zeta_t) \cdot \varphi(\zeta)^\top \varphi(\zeta) \Big] & \text{if } \rho_n \to 0. \end{cases} \tag{9}$$

Then there exists a deterministic matrix $\mathbf{R}_* \in \mathbb{R}^{d \times d}$ and a sequence of orthogonal matrices $\mathbf{W} \in O(d)$ such that, given $\mathbf{z} \in \mathcal{Z}$, for all $\mathbf{y} \in \mathbb{R}^d$ and for any fixed $i \in [n]$ and $t \in [T]$,

$$\mathbb{P}\Big( n^{1/2} (\hat{\mathbf{Y}}_i^{(t)} \tilde{\mathbf{W}} - \tilde{\mathbf{Y}}_i^{(t)}) \mathbf{W} \leq \mathbf{y} \mid \mathbf{Z}_i^{(t)} = \mathbf{z} \Big) \to \Phi\big( \mathbf{y}, \mathbf{R}_* \mathbf{\Sigma}_t(\mathbf{z}) \mathbf{R}_*^\top \big). \tag{10}$$

As in our previous results, the matrices $\mathbf{W}$ and $\mathbf{R}_*$ can be explicitly constructed given knowledge of the underlying function $f$ and distribution $\mathcal{F}$ (again, we defer full details to the supplemental material). Note that as in Proposition 2, both constructed matrices are common to *all* embeddings.

We can now demonstrate one of the key advantages that UASE holds over other embedding methods, namely that *UASE exhibits both cross-sectional and longitudinal stability*, in a sense that we shall now define. We say that two space-time positions $(\mathbf{z}, t)$ and $(\mathbf{z}', t')$ are *exchangeable* if $f(\mathbf{z}, \zeta_t) = f(\mathbf{z}', \zeta_{t'})$ with probability one, where $\zeta = (\zeta_1 | \cdots | \zeta_T) \sim \mathcal{F}$, and that the positions are *exchangeable up to degree* if $f(\mathbf{z}, \zeta_t) = \alpha f(\mathbf{z}', \zeta_{t'})$ for some $\alpha > 0$. Equivalently, $(\mathbf{z}, t)$ and $(\mathbf{z}', t')$ are exchangeable if conditional on $\mathbf{Z}_i^{(t)} = \mathbf{z}$ and $\mathbf{Z}_j^{(t')} = \mathbf{z}'$ the $i$th row of $\mathbf{P}^{(t)}$ and $j$th row of $\mathbf{P}^{(t')}$ are equal with probability one.

**Definition 4.** Given a generic method for dynamic network embedding, with output denoted $(\hat{\mathbf{Z}}_i^{(t)})_{i \in [n]; t \in [T]}$, define the following stability properties:

1. *Cross-sectional stability:* Given exchangeable $(\mathbf{z}, t)$ and $(\mathbf{z}', t)$, $\hat{\mathbf{Z}}_i^{(t)}$ and $\hat{\mathbf{Z}}_j^{(t)}$ are asymptotically equal, with identical error distribution, conditional on $\mathbf{Z}_i^{(t)} = \mathbf{z}$ and $\mathbf{Z}_j^{(t)} = \mathbf{z}'$.

2. *Longitudinal stability:* Given exchangeable $(\mathbf{z}, t)$ and $(\mathbf{z}, t')$, $\hat{\mathbf{Z}}_i^{(t)}$ and $\hat{\mathbf{Z}}_i^{(t')}$ are asymptotically equal, with identical error distribution, conditional on $\mathbf{Z}_i^{(t)} = \mathbf{Z}_i^{(t')} = \mathbf{z}$.

The following result then shows that UASE exhibits both types of stability:

**Corollary 5.** Conditional on $\mathbf{Z}_i^{(t)} = \mathbf{z}$ and $\mathbf{Z}_j^{(t')} = \mathbf{z}'$, the following properties hold:

1. If $(\mathbf{z}, t)$ and $(\mathbf{z}', t')$ are exchangeable then $\hat{\mathbf{Y}}_i^{(t)}$ and $\hat{\mathbf{Y}}_j^{(t')}$ are asymptotically equal, with identical error distribution.

2. If $(\mathbf{z}, t)$ and $(\mathbf{z}', t')$ are exchangeable up to degree then $\hat{\mathbf{Y}}_i^{(t)}$ and $\alpha \hat{\mathbf{Y}}_j^{(t')}$ are asymptotically equal and, under a sparse regime, their error distributions are equal up to scale, satisfying $\mathbf{\Sigma}_t(\mathbf{z}) = \alpha \mathbf{\Sigma}_{t'}(\mathbf{z}')$.

We can gain insight into our theoretical results by applying them in the context of common statistical models. Under the dynamic stochastic block model of Section 2, a pair $(\mathbf{z}, t)$ and $(\mathbf{z}', t')$ are

exchangeable if and only if the corresponding rows of $\mathbf{B}^{(t)}$ and $\mathbf{B}^{(t')}$ are identical. Proposition 3 then predicts that the point cloud obtained via UASE should decompose into a finite number of *Gaussian* clusters, as we observe in Figure 1. Moreover, when combined with Corollary 5 the equality of the fourth rows of $\mathbf{B}^{(1)}$ and $\mathbf{B}^{(2)}$ implies that we should expect *identical* clusters (centre and shape) corresponding to the fourth community at both time points (indicating longitudinal stability) and similarly the equality of the first and second rows of $\mathbf{B}^{(2)}$ tells us that the communities should be indistinguishable at the second time point (indicating cross-sectional stability), behaviours we observe in Figure 1. Under degree-correction [18, 29], Corollary 5 indicates that we should at least observe cross-sectional and longitudinal stability along 'rays' representing communities, a behaviour that is exhibited in Figure 2.

## 4 Comparison

In this section we investigate the stability properties of alternatives to UASE. Table 1 shows the stability of the three embedding algorithms described in Section 2 and two wider classes of algorithms, described below. For alternatives to UASE, it will be considered sufficient to establish whether an embedding is stable when applied to the Gram matrices $\mathbf{P}^{(1)}, \ldots, \mathbf{P}^{(T)}$. A method found to be unstable in this noise-free condition is not expected to be stable when the matrices are replaced by their noisy observations $\mathbf{A}^{(1)}, \ldots, \mathbf{A}^{(T)}$.

Table 1: Classes of dynamic network embedding algorithms: a brief description, the algorithm complexity and its cross-sectional/longitudinal stability. In the three latter classes, one may replace $\mathbf{A}^{(t)}$ with other matrix representations of the graph (e.g. the normalised Laplacian). Further details in main text.

| Algorithm | Description | Complexity | Stability |
|---|---|---|---|
| UASE [17] | Embed $\mathbf{A} = (\mathbf{A}^{(1)} \mid \cdots \mid \mathbf{A}^{(T)})$ | $\mathrm{O}(dTn^2)$ | Both |
| Omnibus [26] | Embed $\tilde{\mathbf{A}}$; $\tilde{\mathbf{A}}_{s,t} = (\mathbf{A}^{(s)} + \mathbf{A}^{(t)})/2$ | $\mathrm{O}(\tilde{d}T^2n^2)$ | Longitudinal |
| Independent | Embed $\mathbf{A}^{(t)}$ | $\mathrm{O}(\sum_t d_t n^2)$ | Cross-sectional |
| Separate embedding [40, 8, 4, 32, 37, 19] | Embed $\bar{\mathbf{A}}^{(t)} = \sum_k w_k \mathbf{A}^{(t-k)}$ | | Neither |
| Joint embedding [27, 7, 54, 5, 28] | $\arg\min_{\hat{\mathbf{Y}}^{(1)},\ldots,\hat{\mathbf{Y}}^{(T)}} \alpha \sum_t \mathrm{CS}(\hat{\mathbf{Y}}^{(t)}, \mathbf{A}^{(t)})$ $+ (1-\alpha) \sum_t \mathrm{CT}(\hat{\mathbf{Y}}^{(t)}, \hat{\mathbf{Y}}^{(t+1)})$ | | Neither |

The omnibus method computes the spectral embedding of the matrix $\tilde{\mathbf{A}} \in \{0, 1\}^{nT \times nT}$ where the block $\tilde{\mathbf{A}}_{s,t} \in \mathbb{R}^{n \times n}$ is given by $(\mathbf{A}^{(k)} + \mathbf{A}^{(\ell)})/2 \in \mathbb{R}^{n \times n}$ and we denote by $\tilde{\mathbf{P}} \in [0, 1]^{nT \times nT}$ the noise-free counterpart of $\tilde{\mathbf{A}}$. If $(\mathbf{z}, t)$ and $(\mathbf{z}, t')$ are exchangeable, then, conditional on $\mathbf{Z}_i^{(t)} = \mathbf{Z}_i^{(t')} = \mathbf{z}$, the rows $\tilde{\mathbf{P}}_{n(t-1)+i}$ and $\tilde{\mathbf{P}}_{n(t'-1)+i}$ are equal. However, if $(\mathbf{z}, t)$ and $(\mathbf{z}', t)$ are exchangeable, then, in general, $\tilde{\mathbf{P}}_{n(t-1)+i} \neq \tilde{\mathbf{P}}_{n(t-1)+j}$ when $\mathbf{Z}_i^{(t)} = \mathbf{z}$ and $\mathbf{Z}_j^{(t)} = \mathbf{z}'$. One can demonstrate that two rows of $\tilde{\mathbf{P}}$ are equal if and only if the corresponding nodes' embeddings are too. Therefore, omnibus embedding provides longitudinal but not cross-sectional stability.

Independent adjacency spectral embedding computes the spectral embeddings of the matrices $\mathbf{A}^{(t)} \in \{0, 1\}^{n \times n}$. If $(\mathbf{z}, t)$ and $(\mathbf{z}', t')$ are exchangeable, then, conditional on $\mathbf{Z}_i^{(t)} = \mathbf{z}$ and $\mathbf{Z}_j^{(t')} = \mathbf{z}'$, the rows $\mathbf{P}_i^{(t)}$ and $\mathbf{P}_j^{(t')}$ are equal. If $t = t'$ the embeddings of nodes $i$ and $j$ are equal, however, embeddings between different graphs are subject to (possibly indefinite) orthogonal transformations $\mathbf{Q}^{(t)}$ which, if $\mathbf{P}^{(t)} \neq \mathbf{P}^{(t')}$, differ in a non-trivial way (i.e. *beyond* simply reflecting the ambiguity of choosing eigenvectors in the spectral decomposition of $\mathbf{P}^{(t)}$). Therefore, independent adjacency spectral embedding provides cross-sectional but not longitudinal stability. The same arguments extend to other independent embeddings.

Separate embedding covers a collection of embedding techniques separately applied to time-averaged matrices, $\bar{\mathbf{A}}^{(t)} = \sum_k w_k \mathbf{A}^{(t-k)}$ where $w_k$ are non-negative weights, and $\mathbf{A}^{(t)}$ may be replaced by

another matrix representation of the graph such as the normalised Laplacian. The weights may be constant, e.g. $w_k = 1/t$ for all $k$ [40, 4], exponential forgetting factors $w_k = (1 - \lambda)^k$ [8, 19], chosen to produce a sliding window [32], based on a time series model [37], and more. In general, temporal smoothing results in two nodes behaving identically at time $t$ being embedded differently if their past or future behaviours differ, whereas the act of embedding the matrices separately will result in the same issues of alignment encountered in independent adjacency spectral embedding. Therefore, those methods can have neither cross-sectional nor longitudinal stability, except in special cases where one can contrive to have one but not other (e.g., $w_0 = 1$ and $w_k = 0$, reducing to independent embedding).

Joint embedding techniques generally aim to find an embedding to trade-off two costs: a 'snapshot cost' (CS) measuring the goodness-of-fit to the observed $\mathbf{A}^{(t)}$, and a 'temporal cost' (CT) penalising change over time. These are then combined into a single objective function, for example, as [28] (where we have replaced the normalised Laplacian by the adjacency matrix):

$$\underset{\breve{\mathbf{A}}^{(1)}, \ldots, \breve{\mathbf{A}}^{(T)}}{\arg\min} \ \alpha \sum_{t=1}^{T} \|\mathbf{A}^{(t)} - \breve{\mathbf{A}}^{(t)}\|_F^2 + (1 - \alpha) \sum_{t=1}^{T-1} \|\breve{\mathbf{A}}^{(t)} - \breve{\mathbf{A}}^{(t+1)}\|_F^2, \tag{11}$$

where $\alpha \in [0, 1]$, subject to a low rank constraint on $\breve{\mathbf{A}}^{(1)}, \ldots, \breve{\mathbf{A}}^{(T)}$, where $\hat{\mathbf{Y}}^{(t)}$ is the spectral embedding of $\breve{\mathbf{A}}^{(t)}$. It is easy to see that a change affecting only a fraction of the nodes will result in a change of all node embeddings, precluding both cross-sectional and longitudinal stability (again apart from contrived cases, e.g. when $\alpha = 1$).

For the complexity calculations, we assume a dense regime in which the $k$-truncated singular value decomposition of an $m$-by-$n$ matrix is $O(kmn)$ [13]. Independent embedding is more efficient than UASE, on account of $\sum_t d_t \leq dT$, but UASE is more efficient than omnibus embedding, because of the linear versus quadratic growth in $T$, while $\tilde{d} = \text{rank}(\tilde{\mathbf{P}})$ is often larger than $d$.

# 5 Real data

The Lyon primary school data set shows the social interactions at a French primary school over two days in October 2009 [44]. The school consisted of 10 teachers and 241 students from five school years, each year divided into two classes. Face-to-face interactions were detected when radio-frequency identification devices worn by participants (10 teachers and 232 students gave consent to be included in the experiment) were in close proximity over an interval of 20 seconds and recorded as a pair of anonymous identifiers together with a timestamp. The data are available for download from the Network Repository website[1] [34].

A time series of networks was created by binning the data into hour-long windows over the two days, from 08:00 to 18:00 each day. If at least one interaction was observed between two people in a particular time window, an edge was created to connect the two nodes in the corresponding network. This results in a time series of graphs $\mathbf{A}^{(1)}, \ldots, \mathbf{A}^{(20)}$ each with $n = 242$ nodes. Where a node is not active in a given time window, it is still included in the graph as an isolated node. This is compatible with the theory and method, and the node is embedded to the zero vector at that time point.

Given the unfolded adjacency matrix $\mathbf{A} = (\mathbf{A}^{(1)} | \cdots | \mathbf{A}^{(20)})$, an estimated embedding dimension $\hat{d} = 10$ was obtained using profile likelihood [55] and we construct the embeddings $\hat{\mathbf{Y}}^{(1)}, \ldots, \hat{\mathbf{Y}}^{(20)} \in \mathbb{R}^{n \times 10}$, taking approximately five seconds on a 2017 MacBook Pro. Figure 2 shows the first two dimensions of this embedding to visualise some of the structure in the data. Similar plots and discussion for both individual spectral embedding and omnibus embedding are given in the Appendix.

From this plot, we observe clustering of students in the same school class. For time windows corresponding to classroom time, for example, 09:00–10:00 and 15:00–16:00, the embedding forms rays of points in 10-dimensional space, with each ray broadly corresponding to a single school class. This is to be expected under a degree-corrected stochastic block model, and the distance along the ray is a measure of the node's activity level [25, 30, 38]. However, not all time windows exhibit this structure, for example, the different classes mix more during lunchtimes (time windows 12:00–13:00 and 13:00–14:00).

---

[1] https://networkrepository.com

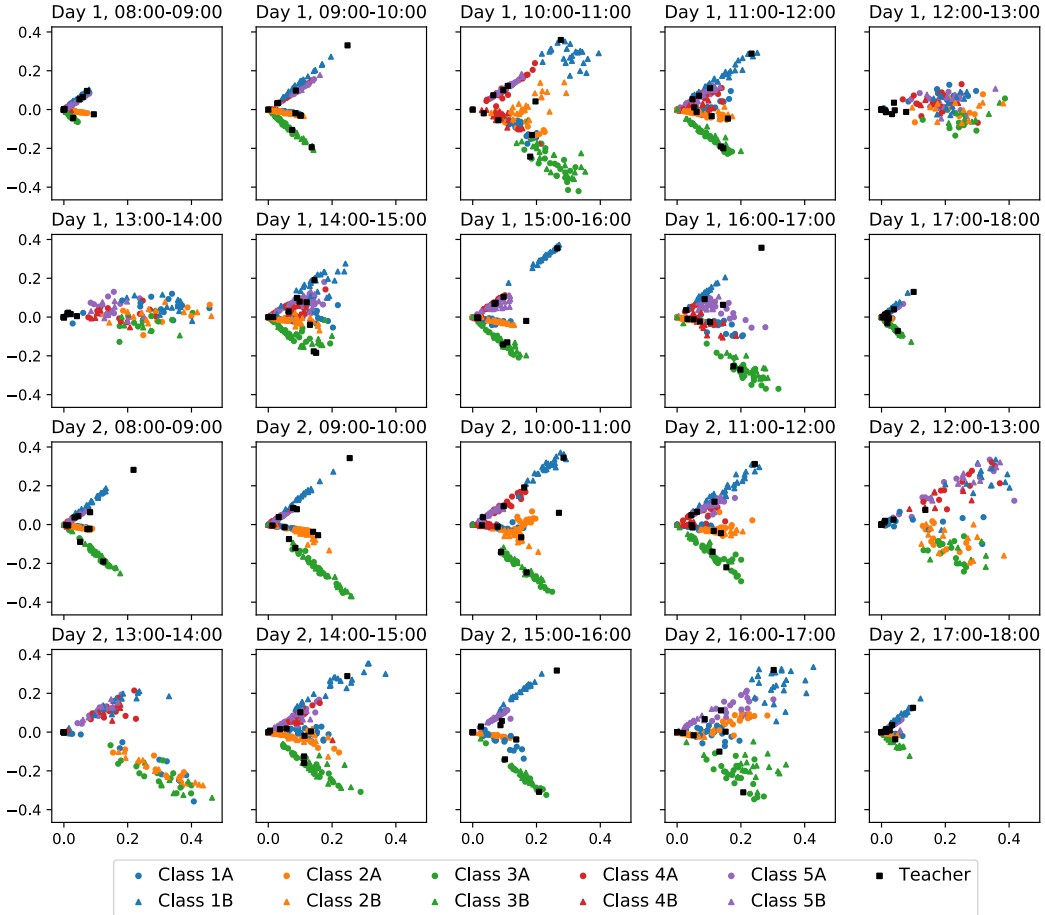

Figure 2: First two dimensions of the embeddings $\hat{\mathbf{Y}}^{(1)}, \ldots, \hat{\mathbf{Y}}^{(20)}$ of the unfolded adjacency matrix $\mathbf{A} = (\mathbf{A}^{(1)} | \cdots | \mathbf{A}^{(20)})$. The colours indicate different school years while the marker type distinguish the two school classes within each year.

## 5.1 Clustering

Following recommendations regarding community detection under a degree-corrected stochastic block model [38], we analyse UASE using spherical coordinates $\mathbf{\Theta}^{(t)} \in [0, 2\pi)^{n \times 9}$, for $t \in [T]$. Since UASE demonstrates cross-sectional and longitudinal stability, we can combine the embeddings into a single point cloud $\mathbf{\Theta} = (\mathbf{\Theta}^{(1)\top} | \cdots | \mathbf{\Theta}^{(T)\top})^{\top} \in \mathbb{R}^{nT \times 9}$ where each point represents a student or teacher in a particular time window. This allows us to detect people returning to a previous behaviour in the dynamic network. We fit a Gaussian mixture model with varying covariance matrices to the non-zero points in $\mathbf{\Theta}$ with 20–50 clusters increasing in increments of 5, with 50 random initialisations, taking approximately five minutes on a 2017 MacBook Pro. Using the Bayesian Information Criterion, we select the best fitting model (30 clusters) and assign the maximum a posteriori Gaussian cluster membership to each student in each time window.

Figure 3 shows how students in the ten classes move between these clusters over time. Each class has one or two clusters unique to it, for example, the majority of students in class 1A spend their classroom time (as opposed to break time) assigned to cluster 1 or cluster 25. This highlights the importance of longitudinal stability in UASE, as we are detecting points in the embedding returning to some part of latent space.

There are also instances of multiple school classes being assigned the same cluster at the same time period, for example, on the morning of day 1, classes 5A and 5B are mainly in cluster 28 suggesting

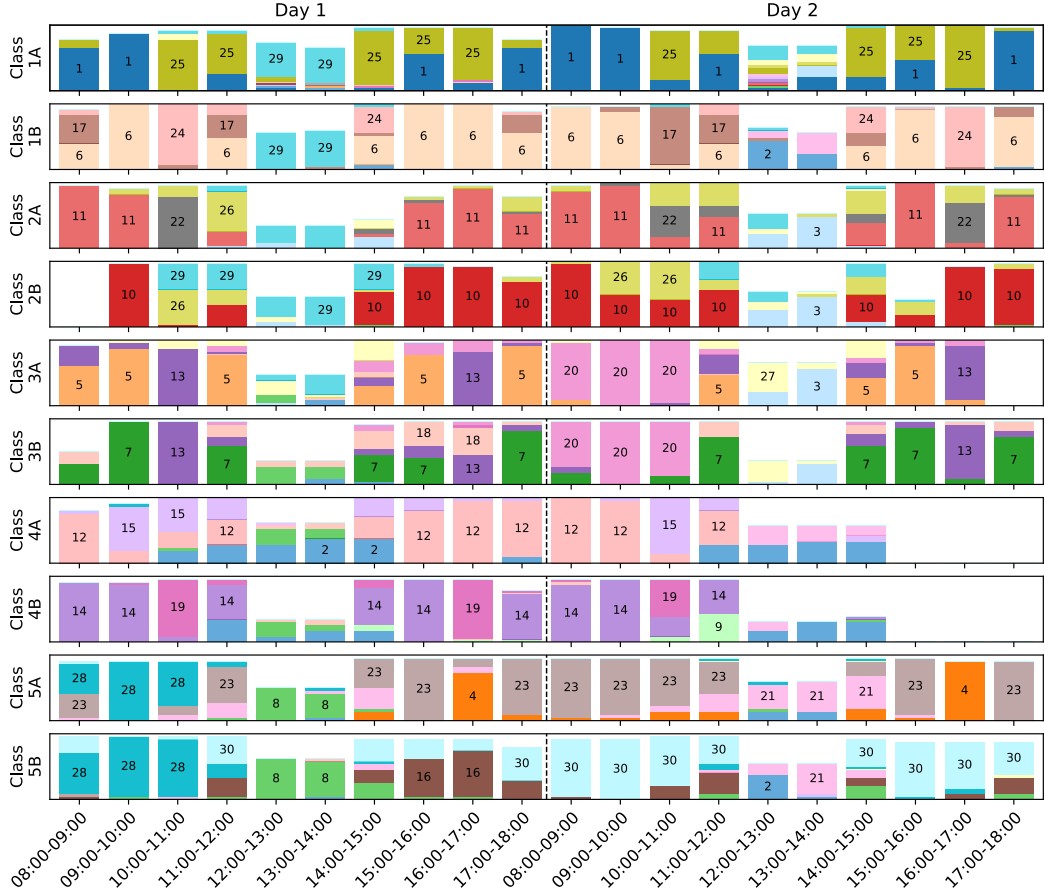

Figure 3: Bar chart showing the Gaussian cluster assignment of school classes over time. The height of each coloured bar represents the proportion of students, in that class and at that time, assigned to the corresponding Gaussian cluster, the total available height representing 100%. If the coloured bars do not sum to the full available height, the difference represents the proportion of inactive students. For legibility, only bars representing over 35% of the class are labelled with the cluster number.

they are having a joint lesson, and we see this behaviour again on day 2 with classes 3A and 3B in cluster 20. In the lunchtime periods, particularly on day 1, the younger students (classes 1A–2B) mingle to form a larger cluster, as do the older students (classes 4A–5B), potentially explained by the cafeteria needing two sittings for lunch for space reasons [44]. This highlights the importance of cross-sectional stability in UASE, as it allows the grouping of nodes behaving similarly in a specific time window, irrespective of their potentially different past and future behaviours.

## 6 Conclusion

We prove that an existing procedure, UASE, allows dynamic network embedding with longitudinal and cross-sectional stability guarantees. These properties make a range of subsequent spatio-temporal analyses possible using 'off-the-shelf' techniques for clustering, time series analysis, classification and more. While our goal is to facilitate exploratory data analysis across a range of scientific disciplines, there could be a privacy concern in the possibility of matching a node's behaviour between two graphs, despite being labelled differently. This could be exploited to compromise someone's identity, for example, across online social networks. We hope to raise awareness of such possibilities, for instance, in the context of network data anonymisation.

## Funding Transparency Statement

Andrew Jones and Patrick Rubin-Delanchy's research was supported by an Alan Turing Institute fellowship. Ian Gallagher's research was supported by an EPSRC PhD studentship.

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
