# A Appendix

Python notebooks producing the figures of this paper are available at `https://github.com/iggallagher/Dynamic-Network-Embedding`.

**Proof of Theorem 1**

Given the factorisation

$$f(\mathbf{x}, \mathbf{y}) = \sum_{i=1}^{D} \lambda_i u_i(\mathbf{x}) u_i(\mathbf{y}), \tag{12}$$

define a map $\phi : \mathcal{Z} \to \mathbb{R}^D$ by setting the $i$th coordinate of $\phi(\mathbf{z})$ to be $|\lambda_i|^{1/2} u_i(\mathbf{z})$, so that for any $\mathbf{x}, \mathbf{y} \in \mathcal{Z}$ we have $f(\mathbf{x}, \mathbf{y}) = \phi(\mathbf{x})^\top \mathbf{I}_{p,q} \phi(\mathbf{y})$ (where $\mathbf{I}_{p,q}$ is the diagonal matrix whose entries are the signs of the eigenvalues $\lambda_i$). Let $\mathcal{F}^*$ be the joint distribution on $\mathbb{R}^{TD}$ obtained by first assigning a random vector $\zeta = (\zeta_1 | \cdots | \zeta_T)$ via $\mathcal{F}$ and then applying the map $\phi$ to each of the components $\zeta_t$, and let $\mathcal{F}_1^*, \ldots, \mathcal{F}_T^*$ denote the corresponding marginal distributions on $\mathbb{R}^D$. Given $\xi \sim \mathcal{F}^*$ and $\xi_t \sim \mathcal{F}_t^*$, define the second moment matrices $\Delta = \mathbb{E}[\xi \xi^\top] \in \mathbb{R}^{TD \times TD}$ and $\Delta_t = \mathbb{E}[\xi_t \xi_t^\top] \in \mathbb{R}^{D \times D}$, and let $r = \mathrm{rank}(\Delta)$ and $r_t = \mathrm{rank}(\Delta_t)$.

Let $\mathbf{M} \in \mathbb{R}^{r \times TD}$ be a matrix whose rows form a basis of $\mathrm{supp}(\mathcal{F}^*)$, and similarly let $\mathbf{N}_t \in \mathbb{R}^{r_t \times D}$ be a matrix whose rows form a basis of $\mathrm{supp}(\mathcal{F}_t^*)$. Let $\mathbf{N} = \mathrm{diag}(\mathbf{N}_1, \ldots, \mathbf{N}_T)$, and define the matrix $\mathbf{\Pi} := \mathbf{M} \mathbf{D} \mathbf{N}^\top \in \mathbb{R}^{r \times (r_1 + \cdots + r_T)}$, where $\mathbf{D} = \mathrm{diag}(\mathbf{I}_{p,q}, \ldots, \mathbf{I}_{p,q})$. Construct the singular value decomposition $\mathbf{\Pi} = \mathbf{U} \mathbf{\Sigma} \mathbf{V}^\top$, with $\mathbf{U} \in \mathrm{O}(r \times d)$, $\mathbf{\Sigma} \in \mathbb{R}^{d \times d}$ and $\mathbf{V} \in \mathrm{O}((r_1 + \cdots + r_T) \times d)$, and $d = \mathrm{rank}(\mathbf{\Pi})$. Writing $\mathbf{V} = (\mathbf{V}_1 | \cdots | \mathbf{V}_T)$, where $\mathbf{V}_t \in \mathbb{R}^{r_t \times d}$ has rank $d_r$, we can then construct the singular value decompositions $\mathbf{V}_t = \mathbf{U}_t \mathbf{\Sigma}_t \mathbf{W}_t^\top$, with $\mathbf{U}_t \in \mathrm{O}(r_t \times d_t)$, $\mathbf{\Sigma}_t \in \mathbb{R}^{d_t \times d_t}$ and $\mathbf{W}_t \in \mathrm{O}(d \times d_t)$, where $d_t = \mathrm{rank}(\mathbf{V}_t)$.

Define $\mathbf{\Lambda} = \mathbf{\Sigma} \mathbf{W} \in \mathbb{R}^{d \times (d_1 + \cdots + d_T)}$, where $\mathbf{W} = (\mathbf{W}_1 \mathbf{\Sigma}_1 | \cdots | \mathbf{W}_T \mathbf{\Sigma}_T)$, and let $L : \mathbb{R}^{TD} \to \mathbb{R}^d$ be the linear map sending the $i$th row of $\mathbf{M}$ to the $i$th row of $\mathbf{U}$, and similarly let $L_t : \mathbb{R}^D \to \mathbb{R}^{d_t}$ be the linear map sending the $i$th row of $\mathbf{N}_t$ to the $i$th row of $\mathbf{U}_t$. Finally, let $\varphi : \mathbb{R}^{Tk} \to \mathbb{R}^d$ and $\varphi_t : \mathbb{R}^k \to \mathbb{R}^{d_t}$ be the maps satisfying $\varphi(\mathbf{z}) = L\big((\phi(\mathbf{z}^{(1)}) | \cdots | \phi(\mathbf{z}^{(T)}))\big)$ and $\varphi_t(\mathbf{z}^{(t)}) = L_t\big(\phi(\mathbf{z}^{(t)})\big)$ for any $\mathbf{z} = (\mathbf{z}^{(1)} | \cdots | \mathbf{z}^{(T)}) \in \mathbb{R}^{Tk}$.

Then, setting $\mathcal{G}$ to be the joint distribution on $\mathbb{R}^d \times \mathbb{R}^{d_1} \times \cdots \mathbb{R}^{d_T}$ obtained by first assigning a random vector $\zeta = (\zeta_1 | \cdots | \zeta_T)$ via $\mathcal{F}$ and then sending this to the tuple $\big(\varphi(\zeta), \varphi_1(\zeta_1), \ldots, \varphi_t(\zeta_t)\big)$ and letting $\mathbf{X}_i = \varphi(\mathbf{Z}_i)$ and $\mathbf{Y}_i^{(t)} = \varphi_t(\mathbf{Z}_i^{(t)})$, we find that $(\mathbf{A}, \mathbf{X}, \mathbf{Y}) \sim \mathrm{MRDPG}(\mathcal{G}, \mathbf{\Lambda})$. $\qquad\square$

**Proof of Proposition 2**

This follows directly from Theorem 1 in [17], which states that there exist sequences of matrices $\mathbf{R}_t = \mathbf{R}_t(n) \in \mathbb{R}^{d_t \times d}$ and $\tilde{\mathbf{W}} \in \mathrm{O}(d)$ such that

$$\|\hat{\mathbf{Y}}^{(t)} - \mathbf{Y}^{(t)} \mathbf{R}_t \tilde{\mathbf{W}}^\top\|_{2 \to \infty} = \mathrm{O}\left(\frac{\log^{1/2}(n)}{\rho^{1/2} n^{1/2}}\right) \tag{13}$$

almost surely, where the matrices $\mathbf{R}_t$ satisfy $\tilde{\mathbf{Y}}^{(t)} = \mathbf{Y}^{(t)} \mathbf{R}_{Y,t}$, and $\tilde{\mathbf{W}} \in \mathrm{O}(d)$ is the solution to the one-mode orthogonal Procrustes problem

$$\tilde{\mathbf{W}} = \underset{\mathbf{Q} \in \mathrm{O}(d)}{\arg \min} \|\mathbf{U}_\mathbf{A} \mathbf{Q} - \mathbf{U}_\mathbf{P}\|_F^2 + \|\mathbf{V}_\mathbf{A} \mathbf{Q} - \mathbf{V}_\mathbf{P}\|_F^2, \tag{14}$$

where $\mathbf{A}$ and $\mathbf{P}$ admit the singular value decompositions $\mathbf{A} = \mathbf{U}_\mathbf{A} \mathbf{\Sigma}_\mathbf{A} \mathbf{V}_\mathbf{A}^\top + \mathbf{U}_{\mathbf{A},\perp} \mathbf{\Sigma}_{\mathbf{A},\perp} \mathbf{V}_{\mathbf{A},\perp}^\top$ and $\mathbf{P} = \mathbf{U}_\mathbf{P} \mathbf{\Sigma}_\mathbf{P} \mathbf{V}_\mathbf{P}^\top$ respectively. Observing that the 2-to-infinity norm of a matrix is known to be equivalent to its maximum Euclidean row norm (and consequently is invariant under orthogonal transformations) gives the desired result. $\qquad\square$

**Proof of Proposition 3**

From the proof of Theorem 1 in [17], we find that

$$n^{1/2}(\hat{\mathbf{Y}}^{(t)} \tilde{\mathbf{W}} - \tilde{\mathbf{Y}}^{(t)}) = n^{1/2}(\mathbf{A}^{(t)} - \mathbf{P}^{(t)}) \mathbf{U}_\mathbf{P} \mathbf{\Sigma}_\mathbf{P}^{-1/2} + n^{1/2} \mathbf{E}, \tag{15}$$

where the residual term $\mathbf{E}$ satisfies $\|n^{1/2}\mathbf{E}\|_{2\to\infty} \to 0$. We can rewrite this as

$$n^{1/2}(\hat{\mathbf{Y}}^{(t)}\tilde{\mathbf{W}} - \tilde{\mathbf{Y}}^{(t)}) = n^{1/2}(\mathbf{A}^{(t)} - \mathbf{P}^{(t)})\mathbf{X}\mathbf{L}\mathbf{\Sigma}_{\mathbf{P}}^{-1} + n^{1/2}\mathbf{E}, \tag{16}$$

where $\mathbf{L} \in \mathrm{GL}(d)$ (the general linear group of invertible $d \times d$ matrices) satisfies $\mathbf{X} = \tilde{\mathbf{X}}\mathbf{L}$, which is known to exist by Proposition 16 of [17].

We begin by showing that there exists a sequence of orthogonal matrices $\mathbf{W} \in \mathrm{O}(d)$ and a fixed matrix $\tilde{\mathbf{L}} \in \mathbb{R}^{d \times d}$ such that $\mathbf{L}\mathbf{W}_* \to \tilde{\mathbf{L}}$, for which we adapt the arguments of Theorem 1 and Corollary 2 of [2].

To begin with, note that the mapping $v \mapsto \mathbf{X}(\mathbf{X}^\top\mathbf{X})^{-1/2}v$ sends the eigenvectors of the matrix $(\mathbf{X}^\top\mathbf{X})^{1/2}\mathbf{\Lambda}\mathbf{Y}^\top\mathbf{Y}\mathbf{\Lambda}^\top(\mathbf{X}^\top\mathbf{X})^{1/2}$ to the left singular vectors of $\mathbf{P}$, since if $v$ is such an eigenvector (with corresponding eigenvalue $\lambda$) then

$$\mathbf{P}\mathbf{P}^\top\mathbf{X}(\mathbf{X}^\top\mathbf{X})^{-1/2}v = \mathbf{X}\mathbf{\Lambda}\mathbf{Y}^\top\mathbf{Y}\mathbf{\Lambda}^\top(\mathbf{X}^\top\mathbf{X})^{1/2}v \tag{17}$$

$$= \mathbf{X}(\mathbf{X}^\top\mathbf{X})^{-1/2}(\mathbf{X}^\top\mathbf{X})^{1/2}\mathbf{\Lambda}\mathbf{Y}^\top\mathbf{Y}\mathbf{\Lambda}^\top(\mathbf{X}^\top\mathbf{X})^{1/2}v \tag{18}$$

$$= \lambda\mathbf{X}(\mathbf{X}^\top\mathbf{X})^{-1/2}v \tag{19}$$

as required. Thus we may write $\mathbf{U}_{\mathbf{P}} = \mathbf{X}(\mathbf{X}^\top\mathbf{X})^{-1/2}\mathbf{V}$, where $\mathbf{V}$ is a matrix of eigenvectors of $(\mathbf{X}^\top\mathbf{X})^{1/2}\mathbf{\Lambda}\mathbf{Y}^\top\mathbf{Y}\mathbf{\Lambda}^\top(\mathbf{X}^\top\mathbf{X})^{1/2}$, and consequently observe that

$$\mathbf{L} = (\mathbf{X}^\top\mathbf{X})^{-1}\mathbf{X}^\top\mathbf{X}_{\mathbf{P}} \tag{20}$$

$$= (\mathbf{X}^\top\mathbf{X})^{-1}\mathbf{X}^\top\mathbf{X}(\mathbf{X}^\top\mathbf{X})^{-1/2}\mathbf{V}\mathbf{\Sigma}_{\mathbf{P}}^{1/2} \tag{21}$$

$$= (\mathbf{X}^\top\mathbf{X})^{-1/2}\mathbf{V}\mathbf{\Sigma}_{\mathbf{P}}^{1/2} \tag{22}$$

$$= \left(\frac{\mathbf{X}^\top\mathbf{X}}{n}\right)^{-1/2}\mathbf{V}\left(\frac{\mathbf{\Sigma}_{\mathbf{P}}}{n}\right)^{1/2}. \tag{23}$$

Let $\mathcal{G}_{\mathrm{X}}$ and $\mathcal{G}_{\mathrm{Y},t}$ denote the marginal distributions of $\mathcal{G}$, and define the second moment matrices $\Delta_{\mathrm{X}} = \mathbb{E}[\xi\xi^\top]$ and $\Delta_{\mathrm{Y},t} = \mathbb{E}[\xi_t\xi_t^\top]$, where $\xi \sim \mathcal{G}_{\mathrm{X}}$ and $\xi_t \sim \mathcal{G}_{\mathrm{Y},t}$, and let $\Delta_{\mathrm{Y}} = \mathrm{diag}(\Delta_{\mathrm{Y},1},\ldots,\Delta_{\mathrm{Y},T})$. Then the law of large numbers tells us that the first and last terms in (23) converge to $(\rho_n\Delta_{\mathrm{X}})^{-1/2}$ and $(\rho_n\tilde{\mathbf{\Sigma}})^{1/2}$ respectively, where $\tilde{\mathbf{\Sigma}}$ is the diagonal matrix whose entries are the square roots of the eigenvalues of $\Delta_{\mathrm{X}}^{1/2}\mathbf{\Lambda}\Delta_{\mathrm{Y}}\mathbf{\Lambda}^\top\Delta_{\mathrm{X}}^{1/2}$ (see, for example, Proposition 7 of [17]). Note that $\mathbf{V}$ is also the matrix of eigenvectors of

$$\left(\frac{\mathbf{X}^\top\mathbf{X}}{n}\right)^{1/2}\left(\frac{\mathbf{\Lambda}\mathbf{Y}^\top\mathbf{Y}\mathbf{\Lambda}^\top}{n}\right)\left(\frac{\mathbf{X}^\top\mathbf{X}}{n}\right)^{1/2} \tag{24}$$

which converges to $\Delta_{\mathrm{X}}^{1/2}\mathbf{\Lambda}\Delta_{\mathrm{Y}}\mathbf{\Lambda}^\top\Delta_{\mathrm{X}}^{1/2}$ by the law of large numbers. Consequently, for each distinct eigenvalue of $\Delta_{\mathrm{X}}^{1/2}\mathbf{\Lambda}\Delta_{\mathrm{Y}}\mathbf{\Lambda}^\top\Delta_{\mathrm{X}}^{1/2}$ we may apply the Davis-Kahan theorem to find that the principal angles between the resulting eigenspace and the subspace spanned by the corresponding columns of $\mathbf{V}$ vanish, and thus $\mathbf{V}$ converges to $\tilde{\mathbf{V}}$ up to some block-orthogonal transformation $\mathbf{W} \in \mathrm{O}(d)$, where $\tilde{\mathbf{V}}$ is a fixed matrix of eigenvectors of $\Delta_{\mathrm{X}}^{1/2}\mathbf{\Lambda}\Delta_{\mathrm{Y}}\mathbf{\Lambda}^\top\Delta_{\mathrm{X}}^{1/2}$. Since $\mathbf{W}$ by definition commutes with $\tilde{\mathbf{\Sigma}}$, we find that

$$\mathbf{L}\mathbf{W} \to \Delta_{\mathrm{X}}^{-1/2}\tilde{\mathbf{V}}\tilde{\mathbf{\Sigma}}^{1/2} \tag{25}$$

as required.

Multiplying (16) by $\mathbf{W}$, we find that

$$n^{1/2}(\hat{\mathbf{Y}}_i^{(t)}\tilde{\mathbf{W}} - \tilde{\mathbf{Y}}_i^{(t)})\mathbf{W} \approx n\rho_n\left[\frac{1}{n^{1/2}\rho_n}(\mathbf{A}^{(t)} - \mathbf{P}^{(t)})\mathbf{X}\right]_i\mathbf{L}\mathbf{\Sigma}_{\mathbf{P}}^{-1}\mathbf{W} \tag{26}$$

and note that the term $n\rho_n\mathbf{L}\mathbf{\Sigma}_{\mathbf{P}}^{-1}\mathbf{W}$ converges to $\tilde{\mathbf{L}}\tilde{\mathbf{\Sigma}}^{-1}$ from our previous discussion. Moreover,

$$\left[\frac{1}{n^{1/2}\rho_n}(\mathbf{A}^{(t)} - \mathbf{P}^{(t)})\mathbf{X}\right]_i = \frac{1}{n^{1/2}\rho_n}\sum_{j=1}^{n}(\mathbf{A}_{ij}^{(t)} - \mathbf{P}_{ij}^{(t)})\mathbf{X}_j \tag{27}$$

$$= \frac{1}{(n\rho_n)^{1/2}}\sum_{j=1}^{n}(\mathbf{A}_{ij}^{(t)} - \mathbf{P}_{ij}^{(t)})\varphi(\mathbf{Z}_j). \tag{28}$$

Conditional on $\mathbf{Z}_i^{(t)} = \mathbf{z}$, we have $\mathbf{P}_{ij}^{(t)} = \rho_n f(\mathbf{z}, \mathbf{Z}_j^{(t)})$, and so the sum in (28) is a scaled sum of $n - 1$ independent, identically distributed zero-mean random variables, each with covariance matrix

$$\mathbb{E}_\zeta \left[ f(\mathbf{z}, \zeta_t) \big( 1 - \rho_n f(\mathbf{z}, \zeta_t) \big) \cdot \varphi(\zeta)^\top \varphi(\zeta) \right] \tag{29}$$

where $\zeta \sim \mathcal{F}$, from which the result follows by setting $\mathbf{R}_* = \tilde{\boldsymbol{\Sigma}}^{-1} \tilde{\mathbf{L}}^\top$ and applying the multivariate versions of the central limit theorem and Slutsky's theorem. $\qquad \square$

**Proof of Corollary 5**

Note that for any $t \in [T]$ the equality $\tilde{\mathbf{X}} \tilde{\mathbf{Y}}^{(t)\top} = \mathbf{P}^{(t)}$ holds, and so $\tilde{\mathbf{Y}}^{(t)} = \mathbf{P}^{(t)} \tilde{\mathbf{X}} (\tilde{\mathbf{X}}^\top \tilde{\mathbf{X}})^{-1}$ (since $\mathbf{P}^{(t)}$ is symmetric). Consequently, for any $i \in [n]$ we find that $\tilde{\mathbf{Y}}_i^{(t)} = (\tilde{\mathbf{X}}^\top \tilde{\mathbf{X}})^{-1} \tilde{\mathbf{X}}^\top \mathbf{P}_i^{(t)}$, where $\mathbf{P}_i^{(t)}$ denotes the $i$th row of $\mathbf{P}^{(t)}$ (again due to symmetry of $\mathbf{P}^{(t)}$). Thus for any exchangeable pair $(\mathbf{z}, t)$ and $(\mathbf{z}', t')$, if $\mathbf{Z}_i^{(t)} = \mathbf{z}$ and $\mathbf{Z}_j^{(t')} = \mathbf{z}'$ then the equality of the rows $\mathbf{P}_i^{(t)}$ and $\mathbf{P}_j^{(t')}$ implies the equality of $\tilde{\mathbf{Y}}_i^{(t)}$ and $\tilde{\mathbf{Y}}_j^{(t')}$. Moreover, from the definition of exchangeability it is clear that the matrices $\boldsymbol{\Sigma}_t(\mathbf{z})$ and $\boldsymbol{\Sigma}_{t'}(\mathbf{z}')$ present in the limiting distributions for $\hat{\mathbf{Y}}_i^{(t)}$ and $\hat{\mathbf{Y}}_j^{(t')}$ in Proposition 3 are equal, and since the matrices $\mathbf{W}$, $\tilde{\mathbf{W}}$ and $\mathbf{R}_*$ are independent of $t$ and $t'$ equality of the full covariance matrices follows.

An analogous argument holds in the case that $(\mathbf{z}, t)$ and $(\mathbf{z}', t')$ are exchangeable up to degree. $\qquad \square$

**Lyon primary school data: other embeddings**

In this section, we show the spectral embeddings for the Lyon primary school data for two other embedding algorithms: independent adjacency spectral embedding and omnibus embedding.

First, for the adjacency matrices $\mathbf{A}^{(1)}, \ldots, \mathbf{A}^{(20)}$, we construct the adjacency spectral embeddings $\hat{\mathbf{Y}}^{(1)}, \ldots, \hat{\mathbf{Y}}^{(20)} \in \mathbb{R}^{n \times 10}$, using the same dimension as the equivalent UASE for this data. This takes approximately 0.2 seconds on a 2017 MacBook Pro. Figure 4 shows the first two dimensions of this embedding to visualise some of the structure in the data.

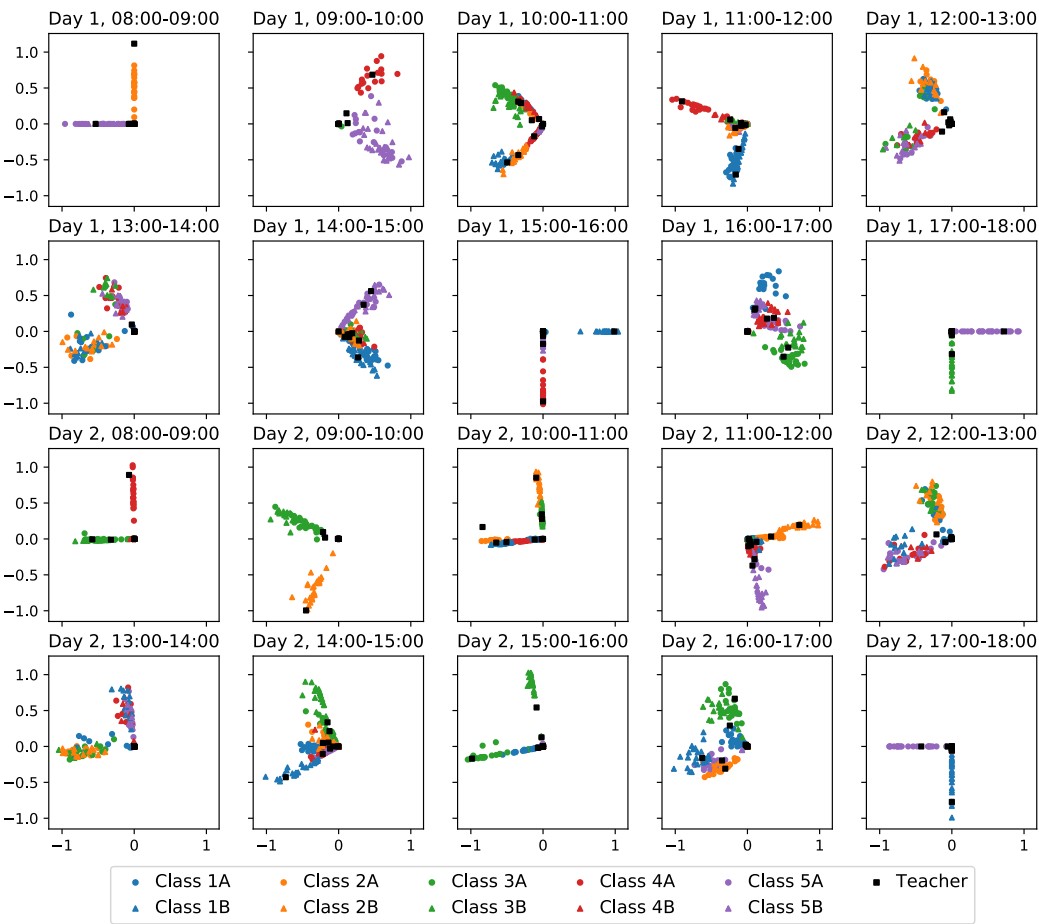

Figure 4: First two dimensions of the embeddings $\hat{\mathbf{Y}}^{(1)}, \ldots, \hat{\mathbf{Y}}^{(20)}$ of the adjacency matrices $\mathbf{A}^{(1)}, \ldots, \mathbf{A}^{(20)}$. The colours indicate different school years while the marker type distinguishes the two school classes within each year.

From this plot, we see that there is no longitudinal stability using individual adjacency spectral embedding. The classes shown in the leading two dimensions are not the same between time periods. Therefore, we cannot make any inference about how the behaviour of students changes over time.

Secondly, we construct the omnibus matrix $\tilde{\mathbf{A}} \in \mathbb{R}^{nT \times nT}$, where the $n$-by-$n$ block of the matrix corresponding to times $s, t$ is given by $\tilde{\mathbf{A}}_{s,t} = (\mathbf{A}^{(s)} + \mathbf{A}^{(t)})/2 \in \mathbb{R}^{n \times n}$. We construct the spectral embedding of $\tilde{\mathbf{A}}$ into $\tilde{d} = 10$ dimensions (as with the other embeddings), to obtain $\hat{\mathbf{Y}} \in \mathbb{R}^{nT \times 10}$. This is divided to get the omnibus embedding for each time period, $\hat{\mathbf{Y}} = (\hat{\mathbf{Y}}^{(1)} | \cdots | \hat{\mathbf{Y}}^{(20)})$. This takes approximately 30 seconds on a 2017 MacBook Pro. Figure 5 shows the first two dimensions of this embedding to visualise some of the structure in the data.

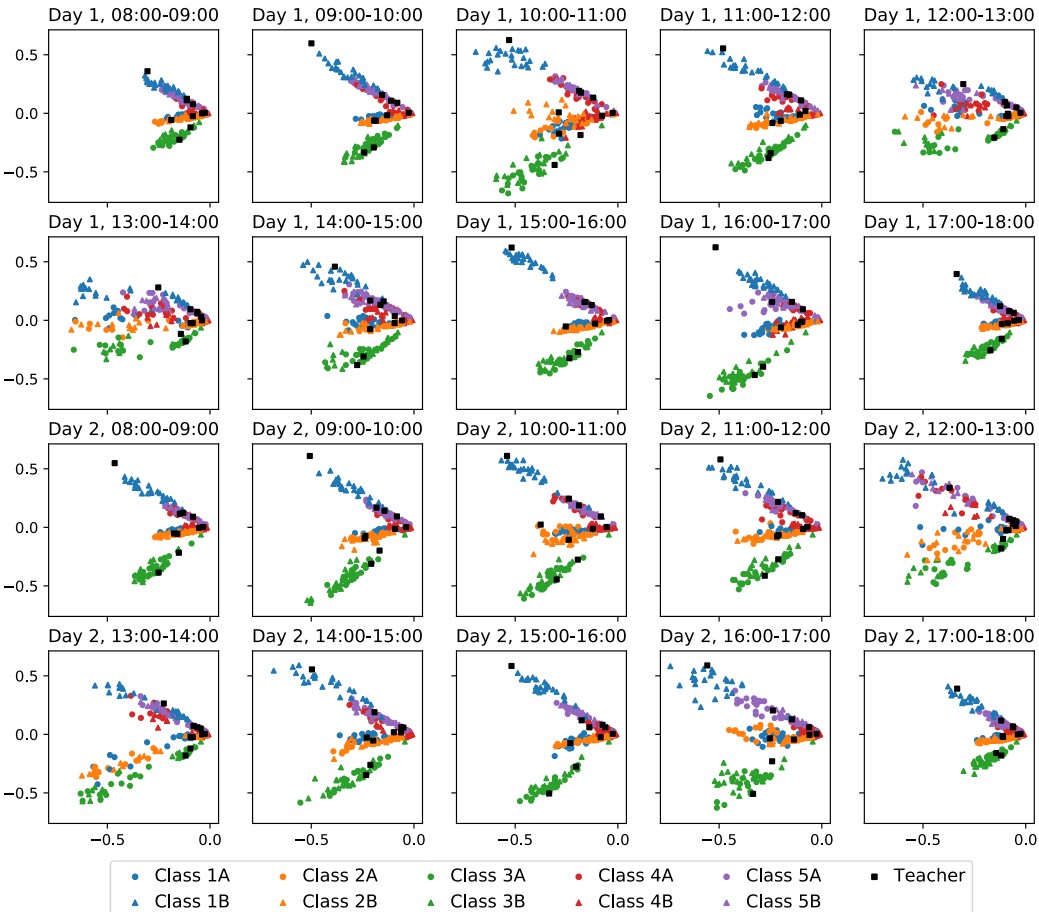

Figure 5: First two dimensions of the embeddings $\hat{\mathbf{Y}}^{(1)}, \ldots, \hat{\mathbf{Y}}^{(20)}$ of the omnibus matrix $\tilde{\mathbf{A}}$. The colours indicate different school years while the marker type distinguish the two school classes within each year.

The rough temporal alignment of the point clouds shows the benefits of omnibus embedding in providing longitudinal stability. However, the lunchtime periods (days 1 and 2, 12:00-14:00) best demonstrate the effect of a lack of cross-sectional stability. Some of the teachers are embedded among what we believe are largely spurious clusters of students during these lunch periods. Empirically, teachers interact much less frequently with students in these periods. Moreover, such a sharp classroom-wise clustering at lunchtime is not present in the independent embedding and seems inconsistent with personal experience. We believe these effects are due to the averaging of past and future behaviours inherent in omnibus embedding, which causes cross-sectional instability.

**Lyon primary school data: classification**

An alternative use of UASE is to analyse the trajectory of each node through time, in embedded space. Because of the longitudinal stability of UASE, standard multivariate time series analysis techniques can be used to detect trends and seasonal behaviour. These, in turn, could enable latent position forecasting and, from this, link prediction. As background, in such analyses, the time series model is usually incorporated in the embedding process, for example, via a Markov model for the communities [5, 12, 54], or a seasonal autoregressive integrated process on the adjacency matrices [37].

In this section, we instead consider the task of time series classification. Given the trajectory for each student, the goal is to predict their school class. Given a arbitrary time series, this is often done using dynamic time warping [42] or convolutional neural networks [53] to allow for misalignment in

time. However, in this simple example with fixed classroom times, we simply fit a random forest classifier with 100 trees to the concatenation of the spectral embeddings $(\Theta^{(1)}|\cdots|\Theta^{(T)}) \in \mathbb{R}^{n \times 9T}$, each using five randomly selected features. The 10-fold cross-validation accuracy is $0.983 \pm 0.035$, taking approximately two seconds on a 2017 MacBook Pro.

When classifying a time series in this way, auto-correlation makes feature importance harder to measure. Nevertheless, certain features appear *not* to be important for classification, in particular, we find that the two lunchtime periods (time windows 12:00-13:00 and 13:00-14:00) are not useful, confirming what is shown by those spectral embeddings in Figure 2.

Using this classifier, we can predict the class of the 10 teachers. While we do not expect them to behave exactly as the students, we might hope to match a teacher to their class. Figure 6 shows the proportion of random forest trees classifying each teacher to each school class. Without truth data it is impossible to know if these labels are correct, but the most likely classification assigns exactly one teacher to each class.

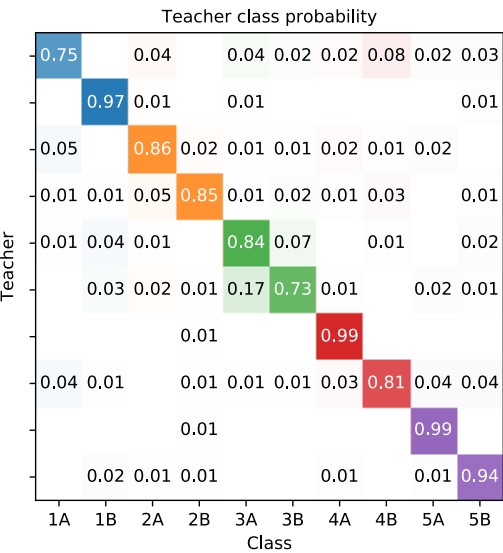

Figure 6: Heat map showing the proportion of random forest trees assigning the spectral embedding trajectory for each teacher to the ten school classes 1A–5B. The colour represents the school class, matching the colours used in Figure 2, where larger proportions are more opaque. Missing values represent a proportion of 0.