# OpenReview forum: "Spectral embedding for dynamic networks with stability guarantees"
_NeurIPS.cc/2021/Conference — NeurIPS 2021 Poster_

### Official Review · Reviewer_irCJ · 2021-07-15

**Rating:** 6
**Confidence:** 3

**Summary:**

The paper analyses several consistency (stability) properties of the unfolded adjacency embedding (UASE), a spectral embedding method for dynamically evolving graphs. Specifically it is shown that UASE yields cross-sectionally and longitudinally stable embeddings. This is contrasted with other (spectral) embedding methods which generally do not achieve these properties.

**Limitations And Societal Impact:**

The discussion on the limitations of the methods appear to be very short. In particular the assumption of an underlying low-rank distribution and of the sparsity level are not discussed in sufficient detail in my opinion.

**Main Review:**

The paper mainly focuses on the analysis of a particular spectral embedding method, UASE, and establishes some important properties of this spectral embedding. The obtained results are overall interesting and appear to have been established in a rigorous way (admittedly I did not check all the fine details).

In my view, the strongest contribution of this paper is a theoretical one (as encapsulated by the provided theorems); the results are not groundbreaking but significant enough, in my opinion, as the proven stability properties are important for application purposes.
My strongest concerns with the paper are with respect to the presentation of the results and the novelty of this work.

In terms of presentations I found the paper to be of very mixed quality. The introduction, the motivating example and the examples read sufficiently well. However, I found the the "main sections" 3 and 4 to be relatively poorly written and difficult to parse.
A few examples:
- "We can associate to the function f a compact, self-adjoint operator" --> how is this done precisely? Are we effectively assuming that we have a graphon like structure here?
- the acronyms GRDPG, RDPG etc are never defined, \Phi is not defined and moreover there is another map \phi appearing before, which may lead to confusion
- the linear maps in Proposition 1, and in particular the dimension d_t are never really explained.

One major point that is not discussed at all is the low-rank assumption on f that is made and its consequences, which seems crucial.
How can this be relaxed? What effects would there be if the assumption is dropped? Would t be sufficient if the eigenvalues are decaying fast enough?

Similarly the assumption on the sparsity of the results would appear to merit more discussion. It appears that the authors consider effectively dense graphs; whereas in most practical applications we would encounter sparse networks.

It would also be of interest to get some insight into how these theoretical results differ from previous results on (low rank network models) in terms of the additional difficulties etc. that are encountered in the proofs. At first glance it seems that the results presented here are reasonably straightforward extensions of results that have been obtain on random dot product graphs etc.

---------------
Update after author response.

As said in my review my strongest concern was with the presentation of some of the results, and the answer provided by the authors appears satisfying to me. I have thus raised my score.


**Time Spent Reviewing:**

4

---

> ### Author Response · Authors · 2021-08-10
> **Response to review**
>
> We would like to thank the reviewer for their time and expertise.
>
> *However, I found the the "main sections" 3 and 4 to be relatively poorly written and difficult to parse.*
>
> This was an issue shared by reviewer BPvN. We describe how we plan to address each specific point below, but to give a high-level overview:
>
> -we will make Section 3 more self-contained, no longer inheriting e.g. acronyms (e.g. GRDPG), model definitions (e.g. MRDPG), constructions (e.g. the self adjoint operator) from other papers.
>
> -we will give some intuition about the mathematical quantities introduced.
>
> -we will make the theoretical novelty of the paper clearer.
>
> To the reviewer comments:
> * *"We can associate to the function f a compact, self-adjoint operator" --> how is this done precisely? Are we effectively assuming that we have a graphon like structure here?*
>
> By extending $f$ to be zero outside of its domain of definition, we may view it as an element of $L^2(\mathbb{R}^k \times \mathbb{R}^k)$.
> We then define the associated operator $A$ via $Ag(x) = \int_{\mathbb{R}^k} f(x,y)g(y)dy$ for $g \in L^2(\mathbb{R}^k)$.  While this construction is given in the reference cited in this section, we are happy to include it to aid clarity.
> On whether we are effectively assuming a graphon structure. Short answer, yes. Longer answer: there is some disagreement in the literature about whether a graphon is a function from $[0,1]^2$ or from $\mathbb{R}^{k \times k}$ more generally. Clearly, we are in the latter situation, although we could reparameterise the $\mathbf{Z}^{(t)}_i$ and $f$ to be in the former. However, a typical reason this reparameterisation is done is to make the simplifying assumption that the latent positions are uniformly distributed on $[0,1]$ --- we do *not* want that as, for example, we do not assume that the $\mathbf{Z}^{(t)}_i$ have the same distribution across time steps.
>
> * *the acronyms GRDPG, RDPG etc are never defined, \Phi is not defined and moreover there is another map \phi appearing before, which may lead to confusion*
>
> As stated earlier, we will make Section 3 more self-contained, no longer inheriting acronyms, model definitions, constructions from other papers. For example, GRDPG will be replaced with generalised random dot product graph, keeping the reference to Rubin-Delanchy et al. (2020). We will add that $\Phi$ is the multivariate Gaussian distribution function (we prefer to keep $\phi$, a common choice of letter for this type of map, e.g. [1]).
>
> [1]Tang, Minh, Daniel L. Sussman, and Carey E. Priebe. "Universally consistent vertex classification for latent positions graphs." The Annals of Statistics 41.3 (2013): 1406-1430.
>
> * *the linear maps in Proposition 1, and in particular the dimension d_t are never really explained.*
>
> The intent of this result is to convey the fact that our dynamic latent position model can in fact be viewed as a multilayer random dot product graph model (in which we have a common "global" set of latent positions across all graphs and a sequence of graph-specific "local" sets of latent positions) which is not immediately obvious from our definition (following reviewer feedback we intend to elaborate on the multilayer model as a prelude to this result).  While definitions of the linear maps involved do appear in the supplementary material, we felt that they did not add extra illumination to the result, as their role is essentially to show that the conversion from "dynamic" latent positions to "global and local" latent positions was well-behaved.  We appreciate that this was not communicated well, however, and will remedy this, either through their inclusion or through a comment similar to the above.  Similarly, the dimensions $d_t$ are to be thought of as the ranks of the underlying probability matrices for each graph (as mentioned above for a dynamic stochastic block model this would be the number of observable communities at each timepoint) --- we will make this clearer:
>
> Line 119 (roughly): We will add that $d_t$ is the approximate rank of $\boldsymbol{A}^{(t)}$, $d$ is the approximate rank of $\boldsymbol{A}$, and that
> "under a dynamic stochastic block model, $d_t$ is the number of distinguishable communities at time $t$, whereas $d$ is the number of distinguishable paths between communities (and the approximate rank of $\boldsymbol{A}$). Hence in Section 2 we have $d=4, d_1 = 4, d_2 = 3$."
>
>
>
> *One major point that is not discussed at all is the low-rank assumption on f that is made and its consequences, which seems crucial. How can this be relaxed? What effects would there be if the assumption is dropped? Would t be sufficient if the eigenvalues are decaying fast enough?*
>
> Reviewer irCJ also had questions about the finite-rank assumption. We will add the following to Section 3:
>
> "
> Several families of functions satisfy the finite rank assumption $D < \infty$, such as the multivariate polynomials [1]. Moreover, several existing statistical models can be written as dynamic latent position network models in which $D < \infty$, including the dynamic mixed membership, degree-corrected, and standard stochastic block models, as we show later. To assume $D < \infty$, more generally, is tantamount to a claim that, for large $n$, $\boldsymbol{A}^{(t)}$ has 'low' approximate rank: an overwhelming proportion of its eigenvalues are close to zero. Large matrices with low approximate rank are routinely encountered across numerous disciplines, and the study [2] provides a hypothesis for this "puzzling" general observation. However, given a real network, we might reject the hypothesis that $f$ has low rank (we must then also reject any type of stochastic block model), for example on the basis of triangle counts [4]. In such a setting, we anticipate that UASE is still consistent and stable, if $d$ is allowed to grow sufficiently slowly with $n$. This claim is supported by asymptotic results for adjacency spectral embedding, for the single graph case, showing convergence in Wasserstein distance under assumptions on eigenvalue decay which can be related to the smoothness of $f$ [3]. However, this notion of convergence is weaker, guaranteeing neither uniform consistency nor a central limit theorem. Even if we could obtain such results for UASE, they would not, for example, have comparable power to those we present here for the case of a dynamic stochastic block model, e.g. to explain the shape of the elliptical clusters in Figure 1.
> "
>
> [1] Rubin-Delanchy, Patrick. "Manifold structure in graph embeddings." Advances in Neural Information Processing Systems 33 (2020).
> [2] Udell, Madeleine, and Alex Townsend. "Why are big data matrices approximately low rank?." SIAM Journal on Mathematics of Data Science 1.1 (2019): 144-160.
> [3] Lei, Jing. "Network representation using graph root distributions." The Annals of Statistics 49.2 (2021): 745-768.
> [4] Seshadhri, C., et al. "The impossibility of low-rank representations for triangle-rich complex networks." Proceedings of the National Academy of Sciences 117.11 (2020): 5631-5637.
>
> *Similarly the assumption on the sparsity of the results would appear to merit more discussion. It appears that the authors consider effectively dense graphs; whereas in most practical applications we would encounter sparse networks.*
>
> There is a typo on line 134 (spotted by reviewer BPvN): the denominator should be $n$ not $n^{1/2}$. We are assuming the average degree grows polylogarithmically in $n$, not linearly --- we would claim this is not "effectively dense"? In any case, we will add the following to Section 3:
>
> "
> In the results to follow, UASE is shown to be consistent and stable in a *uniform sense*, i.e. the maximum error of any position estimate goes to zero, under the assumption that the average network degree grows polylogarithmically in $n$. To achieve this for less than logarithmic growth would be impossible, by *any algorithm*, because it would violate the information-theoretic sparsity limit for perfect community recovery under the stochastic block model [5]. In practice this means that the embedding will have high variance when the graphs have few edges. As mentioned in the introduction, several approaches have been proposed to make single-graph spectral embeddings more robust (e.g. to sparsity), but it is an open and interesting question how to extend them to the dynamic graph setting to achieve both cross-sectional and longitudinal stability.
> "
>
> and discuss robustness (e.g. sparsity, high degree nodes) in the introduction, in line with reviewer BPvN's comments.
>
> [1] Abbe, Emmanuel. "Community detection and stochastic block models: recent developments." The Journal of Machine Learning Research 18.1 (2017): 6446-6531.
>
> *It would also be of interest to get some insight into how these theoretical results differ from previous results on (low rank network models) in terms of the additional difficulties etc. that are encountered in the proofs. At first glance it seems that the results presented here are reasonably straightforward extensions of results that have been obtain on random dot product graphs etc.*
>
> We will make it clear that the novelty is to discover the connection, under low-rank assumptions, between the dynamic latent position model and the multi-layer random dot product graph and, through this, completely unsuspected stability consequences for UASE. Theorems 2 and 3 are relatively straightforward applications of Proposition 1 and the matrix perturbation theory of Jones and Rubin-Delanchy (2021) (plus references therein). Thus we will relabel Proposition 1 to Theorem 1, Theorems 2,3 to Corollaries 2,3, and Corollary 5 to Theorem 5.
>
> *The discussion on the limitations of the methods appear to be very short. In particular the assumption of an underlying low-rank distribution and of the sparsity level are not discussed in sufficient detail in my opinion.*
>
> See above for how we will address both low-rank and sparsity limitations.

---

### Official Review · Reviewer_BPvN · 2021-07-15

**Rating:** 5
**Confidence:** 4

**Summary:**

The paper studies the problem of stability of an embedding of a dynamic network. This corresponds to deriving a time varying vectorial representation for each node such that nodes behaving similarly at a given time have similar positions (cross-sectional stability), and a node behaving similarly across different times has a constant position (longitudinal stability). Assuming  a dynamic random geometric graph model, the authors show for this model that (1) the unfolded adjacency spectral embedding (UASE) procedure satisfies the aforementioned stability properties, and (2) other methods such as omnibus or time-averaged spectral embedding lack one form of stability.

**Limitations And Societal Impact:**

Section 6 does discuss potential societal impact, but I couldn’t find a discussion on the limitations of the results in the paper.

**Main Review:**

The problem studied is an interesting one since dynamic/multi-layer graph related problems appear in many applications. The main issue I have is that the overall presentation of the paper lacks clarity in general, which makes it quite difficult to follow and understand at many places. I have elaborated on this below. Keeping aside this issue for now, I think the technical results in the paper are interesting. However, I am not completely convinced whether they are strong enough for acceptance. The UASE method is but one of several other (much more robust) methods that exist in the literature for handling dynamic or more generally, multi-layer graphs. Therefore, while it is interesting to mathematically analyze UASE, I am currently inclined to believe that this is insufficient for acceptance.

Further remarks:

1.	When the latent position model is defined, it will be good to explain at that point how the SBM and other standard community models are obtained as special cases. This is presently done much later on pg. 5.

2.	The parameter $d$ determines the length of the vector in the embedding, but it should be explained how one can choose this, and what it intuitively corresponds to. For e.g., in the setting where the graphs are identical copies of each other, does it correspond to the number of communities?

3.	There needs to be a notations paragraph that outlines the notation used throughout, this will improve the exposition. Moreover, at some places, it will help to specify the dimensions of the underlying quantities. For e.g. in line 10, $\mathbf{Z}_i \in \mathbb{R}^{k \times T} ?$ for $i=1,\dots,n$?

4.	In line 110, what is GRDPG? In the eigendecomposition of f in eq. (2), there needs to be some explanation on how the smoothness of f is related to the eigenvalues, and some examples of f for which $D < \infty$.

5.	Proposition 1 is particularly difficult to interpret – partially because of unclear notation (what are the sizes of $\mathbf{X}, \mathbf{Y}$ and $\mathbf{\Lambda}$? How are $\mathbf{X}, \mathbf{\Lambda}$ defined?), and partially because no explanation is provided as to why this proposition is useful.

6.	As a general remark, it will help to divide a section using “\paragraph” with an appropriate heading for each paragraph in order to make the exposition clearer for the reader.

7.	In eq. (4), it is unclear to me what the parameter $d$ corresponds to intuitively and how we can choose it, this is related to my comment earlier. In line 134, the sparsity factor has a $\sqrt{n}$ in the denominator but shouldn’t this perhaps be $n$ (in light of the error scaling in theorem 2). The statement of theorem 2 should also have the condition on $\rho_n$. It is also not clear to me how the bound depends on $d$. Clearly $d$ can depend on $n$ so it can’t be hidden within big-Oh. For the same reason, it should also be clarified how $\rho_n$ depends on $d$.

8.	It is important to sketch the proof ideas for Theorems 2 and 3 since these are the main results of the paper. Also, in Corollary 5, the first bullet perhaps has a typo since $t’$ should be $t$? In the second bullet, what does “under a sparse regime mean”?

-------------------------------- Post author response ---------------------------

I thank the authors for their detailed response to my comments and for the clarifications regarding certain technical aspects which I hope will be taken into account in the revision. However, I personally do not find the results in the paper to be strong enough in their current form, and so, I will keep my original score of 5.


**Time Spent Reviewing:**

around 12 hours

---

> ### Author Response · Authors · 2021-08-10
> **Review response**
>
> We would like to thank the reviewer for their time, expertise and very thorough comments.
>
> *The paper lacks clarity in general*
>
> We describe how we plan to address each specific point below, but to give a high-level overview:
> - we will make Section 3 more self-contained, no longer inheriting acronyms (e.g. GRDPG), model definitions (e.g. MRDPG), constructions (e.g. the self adjoint operator) from other papers.
> - we will give some intuition about the mathematical quantities introduced
> - we will make the theoretical novelty of the paper clearer
>
> *The UASE method is but one of several other (much more robust) methods...*
>
> We will take it that the reviewer is referring to the robustness properties of adjacency spectral embedding, which UASE would be expected to inherit. Please let us know if we have misunderstood.
>
> Inspired by this comment, we have over a few days investigated the stability properties of unfolded spectral embedding with other matrix inputs, such as the normalized Laplacian, regularised normalized Laplacian [1], non-backtracking matrices [2], which have shown to be more robust under sparse regimes, degree heterogeneity, and so on. Our expectation was that unfolded spectral embedding would be stable more generally, but we discovered this was false, even with the normalized Laplacian: we lose longitudinal stability if we normalize each matrix independently, and we lose cross-sectional stability we normalize all matrices by the same, average, degree matrix.
>
> In the revision, we will raise these issues of robustness and make it clearer that we are not "selling" UASE; instead we are proving that longitudinal and cross-sectional stability is *possible*, something we don't think was obvious *a priori*, inviting future research into how it can be achieved more robustly.
>
> 1. We will do this on line 40.
>
> 2. Yes. We will address this comment in several places:
>
> Line 119 : We will add that $d_t$ is the approximate rank of $\mathbf{A}^{(t)}$, $d$ is the approximate rank of $\mathbf{A}$, and that "under a dynamic stochastic block model, $d_t$ is the number of distinguishable communities at time $t$, whereas $d$ is the number of distinguishable paths between communities (and the approximate rank of $\mathbf{A}$). Hence in Section 2 we have $d=4, d_1=4, d_2=3$."
>
> Line 128: We will add "In practice, $d$ is estimated as the approximate rank of $\mathbf{A}$. There is no universally optimal way to do this, a point put eloquently in [3]. The method of [4], which uses a profile-likelihood-based analysis of the scree plot, is a popular choice in practice and is easily used within the R package igraph."
>
> 3. We agree that the dimensions need to be made more explicit and some notation clarified. For space considerations, we would prefer to do this as needed, rather than have a notation paragraph, but are more than willing to do this if it aids comprehension. In the given example $\mathbf{Z}_i \in \mathbb{R}^{kT}$ is the concatenation of the row vectors $\mathbf{Z}^{(t)}_i$ - since a similar construction appears more than once (and previously this notation has been used to concatenate matrices) we will make sure to clarify this.
>
> 4. Reviewer irCJ also had questions about the finite-rank assumption. We will add the following to Section 3:
>
> "
> Several families of functions satisfy the finite rank assumption $D < \infty$, such as the multivariate polynomials [5]. Moreover, several existing statistical models can be written as dynamic latent position network models in which $D < \infty$, including the dynamic mixed membership, degree-corrected, and standard stochastic block models, as we show later. To assume $D < \infty$, more generally, is tantamount to a claim that, for large $n$, $\mathbf{A}^{(t)}$ has "low" approximate rank: an overwhelming proportion of its eigenvalues are close to zero. Large matrices with low approximate rank are routinely encountered across numerous disciplines, and the study [6] provides a hypothesis for this "puzzling" general observation. However, given a real network, we might reject the hypothesis that $f$ has low rank (we must then also reject any type of stochastic block model), for example on the basis of triangle counts [7]. In such a setting, we anticipate that UASE is still consistent and stable, if $d$ is allowed to grow sufficiently slowly with $n$. This claim is supported by asymptotic results for adjacency spectral embedding, for the single graph case, showing convergence in Wasserstein distance under assumptions on eigenvalue decay which can be related to the smoothness of $f$ [6]. However, this notion of convergence is weaker, guaranteeing neither uniform consistency nor a central limit theorem (as in Theorems 2 and 3, to follow). Even if we could obtain such results for UASE, they would not, for example, have comparable power to those we present here for the case of a dynamic stochastic block model, e.g. to explain the shape of the elliptical clusters in Figure 1.
> "
>
> 5. We will clarify this result and the accompanying section, by including the definition of a multilayer random dot product graph model and providing additional discussion to the parameters appearing in the Proposition.
> The rows of the $n \times d$ matrix $X$ give a set of "global" latent positions describing node behaviour across the entire time period, while the rows of the $n \times d_t$ matrix $Y^{(t)}_i$ give a set of "local" latent positions describing node behaviour at time $t$. The parameters $d_t$ are the ranks of the underlying probability matrices $\mathbf{P}^{(t)} = X\Lambda^{(t)}Y^{(t)\top}$, while the parameter $d$ is the rank of the concatenation $\mathbf{P} = (\mathbf{P}^{(1)}|\cdots|\mathbf{P}^{(T)})$.
>
> The importance of Proposition 1 is that our model *is* an MRDPG, which is not immediately obvious (the transition from time-varying latent positions to a set of "global" latent positions $X_i$ requires establishing). Given this result, we can adapt existing distributional results to our model, providing a basis from which we can prove the main stability results for the UASE.
>
> While the maps appearing in this result are referred to later Theorem 3 (with explicit constructions given in the supplementary material) we decided not to define them in the main body of the paper as we felt that doing so would not prove illuminating (we merely wished to convey the fact that there was a "well-behaved" manner to construct these latent positions). We appreciate that this leads to a lack of clarity in our exposition, and will remedy this.
>
> 6. We thank the reviewer for bringing this to our attention.
>
> 7. The sparsity factor should indeed have an $n$ in the denominator, this was a typo. As mentioned previously, $d$ is the approximate dimension of the matrix $\mathbf{A}$ (the exact dimension of the underlying matrix $\mathbf{P}$ if $\mathbf{A}$ is known to follow an MRDPG). For our results we have made the simplifying assumption that $d$ is fixed (we appreciate that this was not communicated well, and will rectify this) as our intent was not to draw comparisons between embeddings of different sizes, rather to compare and contrast stability behaviour between embedding types. Indeed, for the CLT to hold true we must have a fixed limit for $d$ as $n \to \infty$. We will clarify this, and are happy to include $d$ in the asymptotic bound of Theorem 2, as our proof keeps track of it. We assume no dependence between $d$ and $\rho_n$.
>
> 8. We feel the request for a sketch proof is a result of a miscommunication on our part on where the theoretical novelty of our work is. We will make it clear that the novelty is to discover the connection, under low-rank assumptions, between the dynamic latent position model and the multi-layer random dot product graph and, through this, completely unsuspected stability consequences for UASE. Theorems 2 and 3 are relatively straightforward applications of Proposition 1 and the matrix perturbation theory of Jones and Rubin-Delanchy (2021) (plus references therein). Thus we will relabel Proposition 1 to Theorem 1, Theorems 2,3 to Corollaries 2,3, and Corollary 5 to Theorem 5. If it aids clarity, we are happy to provide a brief description of the construction of the matrices $W$ and $\tilde{W}$. On $t'$, no: our results are stronger than longitudinal or cross-sectional stability, even allowing matching of one node at time $t$ with another at time $t'$ - this is in fact our critical ethical concern (see conclusion). We will clarify that the first item implies cross-sectional and longitudinal stability (and more). We defined a sparse regime in line 122, but we will add $\rho_n \to 0$ in the Corollary.
>
> [1] Arash A Amini, Aiyou Chen, Peter J Bickel, Elizaveta Levina, et al. Pseudo-likelihood methods for community detection in large sparse networks. The Annals of Statistics, 41(4):2097–2122, 2013.
>
> [2] Krzakala, Florent, et al. "Spectral redemption in clustering sparse networks." Proceedings of the National Academy of Sciences 110.52 (2013): 20935-20940.
>
> [3] Priebe, Carey E., et al. "On a two-truths phenomenon in spectral graph clustering." Proceedings of the National Academy of Sciences 116.13 (2019): 5995-6000.
>
> [4] Zhu, Mu, and Ali Ghodsi. "Automatic dimensionality selection from the scree plot via the use of profile likelihood." Computational Statistics & Data Analysis 51.2 (2006): 918-930.
>
> [5] Rubin-Delanchy, Patrick. "Manifold structure in graph embeddings." Advances in Neural Information Processing Systems 33 (2020).
>
> [6] Udell, Madeleine, and Alex Townsend. "Why are big data matrices approximately low rank?." SIAM Journal on Mathematics of Data Science 1.1 (2019): 144-160.
>
> [7] Seshadhri, C., et al. "The impossibility of low-rank representations for triangle-rich complex networks." Proceedings of the National Academy of Sciences 117.11 (2020): 5631-5637.

---

### Official Review · Reviewer_YyaY · 2021-07-16

**Rating:** 7
**Confidence:** 3

**Summary:**

In this paper the authors study the problem of graph embedding for a series of time-evolving graphs, under the general Random Dot Product Graph framework. They focus on the UASE strategy, comparing it with omnibus and vanilla ASE of each graph, and they provide empirical and theoretical proofs of its capability to preserve the locations of nodes along time, and also to capture relationships between similar nodes in each embedding.

**Limitations And Societal Impact:**

There is a line, stating that the case when T grows presents significant computational challenges. I think this should be further stressed.

**Main Review:**

As stated by the authors, the originality of the paper is not on the UASE method itself, but in the analysis of its properties and the proofs of the results regarding those stability properties. The related work is correctly cited, including some recent papers, and a couple of very relevant reviews. And the difference between this work and the existing literature is clear.
As far as I understand, the results are new.

The paper is also techically sound. The results on the convergence of the embeddings are in line with previous recent results, and provide important properties.
The experimental evaluation is correct and convincing. However, I would have liked to see the performance of omnibus and independent ASE for the Lyon dataset, to further show the strenghts of UASE respect to those two.
Also, I think there should be a comparison on the complexity of the three methods. Basically all three are based on a spectral decomposition, but the dimension of the matrices are different (and in the case of independent ASE, there are T decompositions of $n\times n$ matrices).


The paper is very well written. Both the introduction and the motivating example are very didactical and illustrative. There are some very minor undefined concepts, but in general the quality of the presentation is excellent.

I think the results of this paper are important. It provides a very useful (existing) technique, UASE, now with theoretical guarantees of spatial and temporal consistency.

Minor comments:

In Fig1, the "First two dimensions of the embeddings" are shown, for the tree methods. I couldn't find the original dimension of the embedding used for this example (is it 2 originally?)

(very minor) The MRDPG abbreviation is not defined per se. It is written (multilayer RDPG) two lines above the use of MRDPG.
Also, it would be nice to add a sentence saying that the MRDPG is like a RDPG model but for the concatenaton of the adjacency matrices, or something short like that.


**Time Spent Reviewing:**

5

---

> ### Author Response · Authors · 2021-08-10
> **Review response**
>
> We would like to thank the reviewer for their time and expertise, and all the positive comments.
>
> *In Fig1, the "First two dimensions of the embeddings" are shown, for the tree methods. I couldn't find the original dimension of the embedding used for this example (is it 2 originally?)*
>
> We will clarify the caption for Figure 1 to highlight that the three different approaches to spectral embedding have different dimension; the two leading components of each embedding are shown purely for visualisation. In this example, where the inter-community link probability matrices $\mathbf{B}_1$ and $\mathbf{B}_2$ are known, the true dimension can be computing the rank of the associated $\mathbf{B} $ matrix. For example, for UASE, the matrix $\mathbf{B} = (\mathbf{B}_1 | \mathbf{B}_2)$ has rank 4, while the equivalent matrix for the omnibus embedding has rank 7. In practice, this dimension is unknown and is estimated from the singular values of $\mathbf{A}$ to find the approximate rank. In Section 5 this is done for the Lyon primary school example using profile likelihood estimation [1].
>
> *Also, I think there should be a comparison on the complexity of the three methods.*
>
> We will include a computational complexity analysis of the different spectral embedding techniques in Section 4. This will also be relevant to practitioners interested in using the method for large $T$ (a limitation mentioned in the introduction). In the simplest case, if the complexity of truncated SVD of a $m$-by-$n$ matrix into $k$ dimensions is $O(kmn)$ (there are better approaches, for example, exploiting matrix sparsity), then both individual ASE and UASE have complexity $O(d N^2 T)$, linear in $T$, while omnibus has complexity $O(d N^2 T^2)$.
>
> *However, I would have liked to see the performance of omnibus and independent ASE for the Lyon dataset, to further show the strenghts of UASE respect to those two.*
>
> It was an oversight on our part not to show the performance of other methods such as independent and omnibus embedding on the Lyon primary school data. These experiments are particularly enlightening, for example, highlighting issues caused by the lack of cross-sectional stability in the omnibus embedding. At lunch-break, the students continue to be clustered into classrooms rather than the larger groups. These experiments will be included in the supplementary material.
>
> [1] Zhu, M. and Ghodsi, A. (2006). "Automatic dimensionality selection from the scree plot via the use of profile likelihood." Computational Statistics & Data Analysis, 51(2):918–930.

---

### Official Review · Reviewer_x9sR · 2021-07-17

**Rating:** 7
**Confidence:** 4

**Summary:**

This paper studies the problem of embedding dynamic networks, i.e., providing a node embedding representation for a network that varies with time (a vector representation for each node at each time point). The paper provides a theoretical development showing that an existing method denominated unfolded adjacency spectral embedding (UASE) satisfies desirable stability properties in the embedding across time and across nodes.

**Ethical Concerns:**

no comment.

**Limitations And Societal Impact:**

no comment.

**Main Review:**

The paper nicely motivates the need for stable embeddings for dynamic networks in the introduction. The example in Section 2 is a good illustration of the main concept, though somewhat naive. In particular, the fact that finding independent embeddings for each time point (right column) leads to embeddings that are not stable across time is quite expected, but I think that it is fine to show this as a motivation. I enjoyed the mathematical exposition in Section 3, which is the bulk of the contribution. The result is shown for a fairly general setting, and the discussion after corollary 5 about implications for specific random graph models is helpful.

Section 4, on the other hand, is in my view a weak point of the paper. After having established a stability result for UASE, the authors want to show that other methods are not stable. To do this, they start with a high-level somewhat imprecise categorization of the existing methods into three classes, where the distinction between class 1 and 3 is very thin and the authors justify through an example in (9). The authors then give a few examples of how some of the methods in these categories are not stable. Somewhat ambiguous sentences like “seemingly never providing both [types of stability]” are a bit in contrast with the mathematical precision of Section 3.

The experiments in Section 5 are modest but achieve their objective. Since the paper is not proposing a new methodology but rather a theoretical analysis of stability, there is no need for exhaustive experimentation.

All in all, I deem this paper as a nice analytical contribution to a timely problem.

**Time Spent Reviewing:**

4

---

> ### Author Response · Authors · 2021-08-10
> **Review response**
>
> We would like to thank the reviewer for their time and expertise, and all the positive comments. We describe below how we will address the perceived weaknesses.
>
> *Section 4, on the other hand, is in my view a weak point of the paper. After having established a stability result for UASE, the authors want to show that other methods are not stable. To do this, they start with a high-level somewhat imprecise categorization of the existing methods into three classes,...*
>
> In Section 4, our goal is not to define concrete categories for spectral embedding algorithms, but rather consider similar algorithms together to avoid repetition in our later stability analysis. To address some of this imprecision, we will merge classes 1 and 3 into a single category as the first class can be considered a special case of the third. For example, applying equation 9 to the adjacency matrices with $\alpha = 0$ recovers the independent ASE. We will also address the ambiguity in our statements, for example, replacing "seemingly never providing both [types of stability]" with "to the best of our knowledge never providing both [types of stability]".
>
> Additionally, we will provide a table of existing dynamic graph embedding algorithms, their (possibly implicit) underlying objective function (for example, like the one shown in equation 9), together with their stability properties (cross-sectional, longitudinal).
>
> *The experiments in Section 5 are modest but achieve their objective.*
>
> Thanks. In line with reviewer YyaY's comment, we will extend the experiment in an interesting way. It was an oversight on our part not to show the performance of other methods such as independent and omnibus embedding on the Lyon primary school data. These experiments are particularly enlightening, for example, highlighting issues caused by the lack of cross-sectional stability in the omnibus embedding. At the lunch-break, the students continue to be clustered into classrooms rather than the larger groups. These experiments will be included in the supplementary material.

---

### Author Response · Authors · 2021-08-10
**Thanks**

We would like to thank all the reviewers for their time and expertise. We have responded to each review separately.

---

### Decision · Program_Chairs · 2021-09-27

**Decision:**

Accept (Poster)

**Comment:**

The paper's main contribution is to prove that a specific spectral embedding scheme applied to time-varying graphs, which associates to each node a corresponding time series, ensures in a particular latent position model of random graph two types of stability (cross-sectional and longitudinal). The theory is supplemented by experiments done on a dataset from school in Lyon, France describing student relationships over time. The reviewers generally appreciated the paper's contribution, one review evaluating the results to be slightly below the bar for acceptance.
In agreement with the reviews, I believe the paper's objective to be of interest and to be successfully achieved by the authors' work. While the results may in retrospect appear natural and somehow not very surprising, I would not hold this against the authors, hence my recommendation to accept.